# Quasi-2D Fermi surface in the anomalous superconductor UTe$_2$

A. G. Eaton[1] ✉, T. I. Weinberger [1], N. J. M. Popiel[1], Z. Wu[1], A. J. Hickey [1], A. Cabala [2], J. Pospíšil [2], J. Prokleška[2], T. Haidamak [2], G. Bastien [2], P. Opletal [3], H. Sakai [3], Y. Haga [3], R. Nowell[4], S. M. Benjamin[4], V. Sechovský [2], G. G. Lonzarich[1], F. M. Grosche [1] & M. Vališka [2]

The heavy fermion paramagnet UTe$_2$ exhibits numerous characteristics of spin-triplet superconductivity. Efforts to understand the microscopic details of this exotic superconductivity have been impeded by uncertainty regarding the underlying electronic structure. Here we directly probe the Fermi surface of UTe$_2$ by measuring magnetic quantum oscillations in pristine quality crystals. We find an angular profile of quantum oscillatory frequency and amplitude that is characteristic of a quasi-2D Fermi surface, which we find is well described by two cylindrical Fermi sheets of electron- and hole-type respectively. Additionally, we find that both cylindrical Fermi sheets possess considerable undulation but negligible small-scale corrugation, which may allow for their near-nesting and therefore promote magnetic fluctuations that enhance the triplet pairing mechanism. Importantly, we find no evidence for the presence of any 3D Fermi surface sections. Our results place strong constraints on the possible symmetry of the superconducting order parameter in UTe$_2$.

Conventional phonon-mediated superconductivity involves the pairing of two fermions in a spin-singlet configuration[1] forming a bosonic quasiparticle of total spin $S = 0$. Unconventional superconductors may replace the attractive role played by phonons with a magnetically-mediated pairing interaction, often yielding a $d$-wave symmetry of the orbital wavefunction, but still with overall $S = 0$ for the bound pair[2]. By contrast, the formation of superfluidity in $^3$He involves a triplet pairing configuration with $S = 1$ and an odd-parity $p$-wave symmetry[3]. To date no bulk solid-state analogue of this exotic state of matter has been unequivocally identified, although actinide metals such as UPt$_3$ and UGe$_2$ are among the promising candidates[4,5]. The technological realisation of devices incorporating $p$-wave superconductivity is highly desirable, due to their expected ability to effect coherent quantum information processing[6]. For several years the layered perovskite Sr$_2$RuO$_4$ appeared a likely host of spin-triplet superconductivity[7];

however, recent experimental observations have cast considerable doubt on this interpretation[8].

Since the discovery of unconventional superconductivity in UTe$_2$ in 2019[9], characteristic features of a $p$-wave superconducting state in this material have been reported across numerous physical properties. These include a negligible change in the Knight shift upon cooling through $T_c$ as probed by nuclear magnetic resonance (NMR)[10], high upper critical fields far in excess of the Pauli paramagnetic limit[11], high magnetic field re-entrant superconductivity[12], chiral in-gap states measured by scanning tunneling microscopy (STM)[13], time-reversal symmetry breaking inferred from the development of a finite polar Kerr rotation angle below $T_c$[14,15], multiple point nodes detected from penetration depth measurements indicative of a chiral triplet pairing symmetry[16], anomalous normal fluid properties consistent with Majorana surface arcs[17], and

[1]Cavendish Laboratory, University of Cambridge, JJ Thomson Avenue, Cambridge CB3 0HE, UK. [2]Charles University, Faculty of Mathematics and Physics, Department of Condensed Matter Physics, Ke Karlovu 5, Prague 2 121 16, Czech Republic. [3]Advanced Science Research Center, Japan Atomic Energy Agency, Tokai, Ibaraki 319-1195, Japan. [4]National High Magnetic Field Laboratory, Tallahassee, FL 32310, USA. ✉e-mail: alex.eaton@phy.cam.ac.uk

ferromagnetic fluctuations coexisting with superconductivity measured by muon spin relaxation measurements[18].

Several theoretical studies have sought to provide a microscopic description of how the exotic magnetic and superconducting features manifest in this material[11]. However, an outstanding challenge concerns the determination of the underlying electronic structure, with the question of the geometry and topology of the material's Fermi surface having been the subject of recent debate and speculation[11,19–30]. Two angle-resolved photoemission spectroscopy (ARPES) studies have given contrasting interpretations, with one inferring the presence of multiple small 3D Fermi surface pockets[31], while the other identified spectral features characteristic of a large cylindrical quasi-2D Fermi surface section, along with possible heavy 3D section(s)[32]. A recent de Haas-van Alphen (dHvA) effect study[33] also resolved features representative of a quasi-2D Fermi surface, but with a spectrally dominant low-frequency branch not captured by density functional theory (DFT) calculations, which could be indicative of a 3D Fermi surface pocket. Discerning the dimensionality of the UTe$_2$ Fermi surface is important in order to determine the symmetry of the superconducting order parameter[34].

## Results

Here, we report direct measurements of the UTe$_2$ Fermi surface, probed by magnetic torque measurements of the dHvA effect up to magnetic field strengths of 28 T at various temperatures down to 19 mK. Through measurements of magnetic quantum oscillations as a function of temperature and magnetic field tilt angle, we observe Fermi surface sections with heavy cyclotron effective masses up to 78(2) $m_e$, where $m_e$ is the bare electron mass. We investigated the angular dependence of the quantum oscillations and used the resulting data to perform Fermi surface simulations. Our results indicate that the UTe$_2$ Fermi surface is very well described by two cylindrical Fermi surface sheets of equal volumes and super-elliptical cross sections. In combination, the evolution of quantum oscillatory amplitude and frequency with the tilt angle of the magnetic field, our quantitative analysis of the oscillatory waveform and the contributions from separate Fermi sheets, and the correspondence between the density of states implied from specific heat measurements and our dHvA observations, makes the presence of any 3D Fermi surface pocket(s) very unlikely. We also measured Shubnikov-de Haas (SdH) oscillations in the contactless resistivity (see Supplementary Information). The SdH effect is generally more sensitive than the magnetic torque technique to symmetrical 3D Fermi pockets[35]. However, we find that the SdH response of UTe$_2$ contains no additional frequency components than those probed by dHvA (up to field strengths of 28 T, see Supplementary Information). Furthermore, we find that our quasi-2D Fermi surface model very well describes the relatively small anisotropy of this material's electrical conductivity tensor. In combination these measurements and analysis provide no evidence indicating the presence of 3D sections in the Fermi surface of UTe$_2$, which we instead find to be quasi-2D in nature and composed of two undulating cylindrical sections of hole- and electron-type, respectively.

Samples were grown by the molten salt flux (MSF) technique[36] in excess uranium, to minimise the formation of uranium vacancies (see Methods for details). The MSF technique has been found to produce crystals of exceptionally high quality[36], as demonstrated by specific heat capacity, $C_p$, and electrical resistivity, $\rho$, measurements in Fig. 1. For this batch of crystals on which quantum oscillation studies were performed, we observe a superconducting transition temperature ($T_c$) of 2.1 K and residual resistivity ratios (RRR) of up to 900. The RRR is defined as $\rho$ (300 K)/$\rho_0$, where $\rho_0$ is the residual 0 K resistivity expected for the normal state in the absence of superconductivity, fitted by the dashed line (linear in $T^2$) in Fig. 1b. By comparison, early UTe$_2$ sample generations grown by the chemical vapour transport method exhibited RRR values of ≈ 40 with $T_c$ values of ≈ 1.6 K

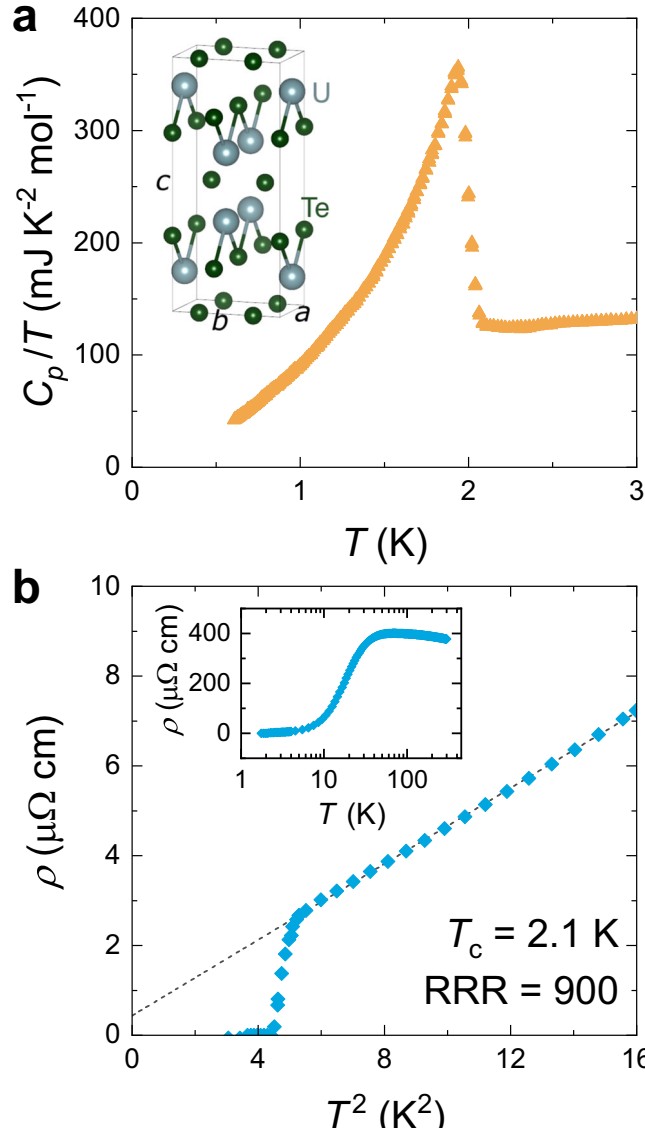

**Fig. 1 | Characterisation of high purity UTe$_2$. a** Specific heat capacity ($C_p$) divided by temperature ($T$) of a UTe$_2$ single crystal measured on warming to 3 K. A single, sharp bulk superconducting transition is exhibited. (Inset) The crystal structure of UTe$_2$. **b** Resistivity ($\rho$) versus temperature squared up to 4 K for current sourced along the $\vec{a}$ direction. A superconducting transition temperature of 2.1 K is observed (defined by zero resistivity, i.e. below the detection limit of 0.01 μΩ cm). A residual resistivity ratio (RRR, defined in the text) of 900 is found, with a residual resistivity $\rho_0 \lesssim 0.5$ μΩ cm, indicative of very high sample purity[36,38]. (Inset) The same dataset as the main panel extended up to 300 K.

(refs. 9,11,37). Optimization of the CVT growth parameters has yielded higher quality crystals with $T_c$ = 2.0 K at a maximal reported RRR of 88 (ref. 38) – however, even these optimized samples exhibit uranium site vacancies of ≈ 0.7%, compared to neglibile vacancies for MSF samples[36,39]. Furthermore, in this study we resolve quantum oscillatory components with frequencies up to 18.5 kT, implying a mean free path of itinerant quasiparticles of at least 1900 Å (see Supplementary Information for calculation), further underlining the pristine quality of this new generation of UTe$_2$ samples. With the advent of these higher quality MSF crystals, it may be necessary to revisit many prior experiments reported on CVT samples – for example, while this manuscript has been undergoing peer review, measurements of the NMR Knight shift[40], Kerr effect[41], thermal transport[42] and muon spectroscopy[43] have been reported on MSF

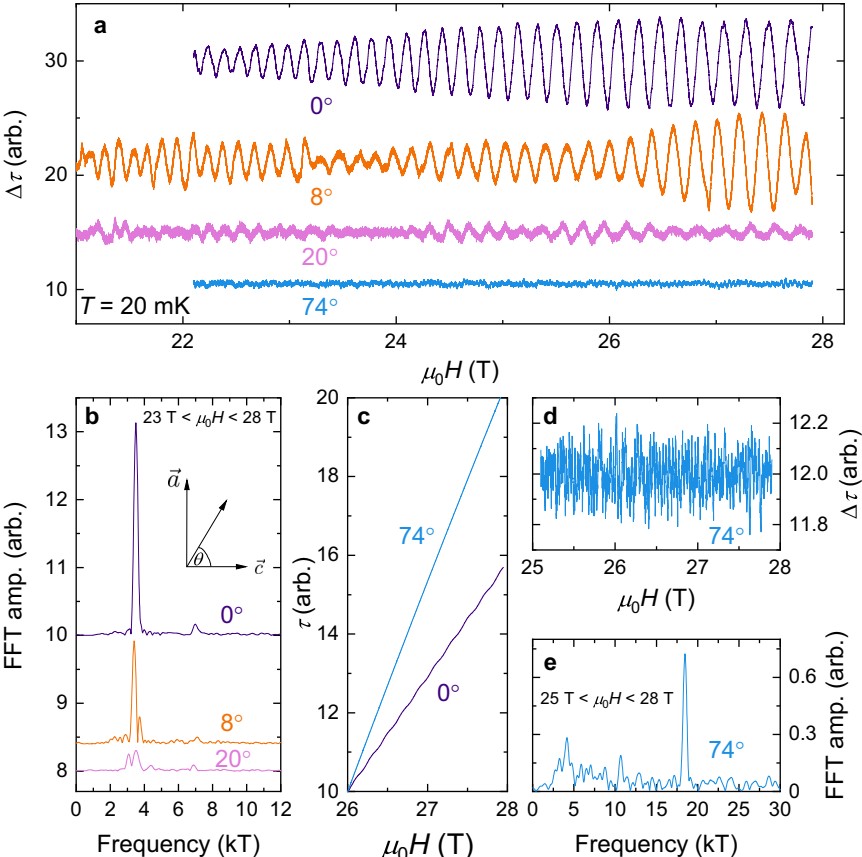

**Fig. 2 | Angular evolution of quantum oscillatory frequencies and amplitude.** **a** Oscillatory component of magnetic torque ($\Delta\tau$) at various angles and **b**, their corresponding Fourier frequency spectra. Angles were calibrated to within 2° of uncertainty. 0° corresponds to magnetic field, $\vec{H}$, applied along the $\vec{c}$ direction; 90° corresponds to field applied along the $\vec{a}$ direction. At 0° a singular frequency peak of high amplitude is observed at $f = 3.5$ kT (with a second harmonic peak at $2f = 7.0$ kT). Upon rotating away from $\vec{c}$ towards $\vec{a}$ this peak splits and the oscillatory amplitude diminishes markedly. **c** Raw magnetic torque signal (without

background subtraction), $\tau$, at 0° and 74°. Both curves have been translated to have the same value of $\tau$ at 26 T for comparison. A clear oscillatory component is visible in the raw torque signal of the 0° data. In comparison, despite the 74° torque being of greater overall magnitude, it is markedly smoother than that at 0°. **d**, $\Delta\tau$ at $\theta = 74°$ and **e**, the corresponding fast Fourier transform (FFT). Despite the very small oscillatory amplitude, a sharp high frequency peak of $f = 18.5$ kT is clearly resolved on top of the background noise profile.

specimens, each of which contrasts with prior interpretations drawn from CVT studies.

Figure 2 shows quantum oscillations measured in the magnetic torque of UTe₂. The oscillatory component of the signal, $\Delta\tau$, was isolated from the background magnetic torque by subtracting a smooth monotonic local polynomial regression fit (see Methods). In panel (a) we find that when a magnetic field, $\vec{H}$, is applied along the $\vec{c}$ direction (defined here as 0°), a monofrequency oscillatory waveform of large amplitude is observed. Upon rotating 8° away from $\vec{c}$ towards the $\vec{a}$ direction, we find here that $\Delta\tau$ incorporates a subtle beat pattern, with the corresponding Fourier spectra indicating the presence of at least one other frequency component in addition to the (now diminished) dominant frequency peak. As the field is tilted further away from the $\vec{c}$ direction we find that this trend progresses: by 20° the oscillatory amplitude has diminished considerably, accompanied by further splitting and broadening of the FFT spectra, while at 74°$\Delta\tau$ appears almost featureless when plotted on this scale.

Figure 2 c compares the raw torque signal, $\tau$, at 0° and 74°. Both curves have been translated (without rescaling) to the same value at 26 T, for ease of comparison. A clear oscillatory component is visible in the 0°$\tau$ curve, whereas the 74° data appear very smooth. Figure 2d, e give a zoomed-in view of $\Delta\tau$ at 74°, in which a very small amplitude (over an order of magnitude smaller than at 0°), high-frequency component of 18.5 kT is clearly present. In the Supplementary Information we perform a quantitative analysis of the 0° waveform, which

reveals the oscillatory contribution of two distinct Fermi surface sections of identical cross-sectional area.

This angular evolution of the dHvA effect – of a singular, relatively low-frequency oscillatory component along a high symmetry direction subsequently evolving under rotation to yield much higher frequencies of smaller amplitude – is characteristic of a cylindrical quasi-2D Fermi surface. This is due to there being negligible smearing of the quantum oscillatory phase when averaging over $k_z$ along the cylindrical axis. Hence, one observes a large quantum oscillatory amplitude for magnetic field oriented in this direction (in this case the $\vec{c}$ direction). Then, as the field is tilted away from the axis of the cylinder, the oscillatory amplitude falls considerably due to phase smearing that increases with the rate of change of the frequency with angle. Furthermore, the oscillatory frequency increases at higher angles because the cross-sectional area of the Fermi surface normal to the magnetic field grows as $\frac{1}{\cos\theta}$ (ref. 35). Thus, our observation of the progression from large amplitude, low-frequency quantum oscillations with field oriented along $\vec{c}$ evolving to small amplitude, high-frequency oscillations for field applied close to $\vec{a}$ is strongly indicative of UTe₂ possessing cylindrical quasi-2D Fermi surface sections, axially collinear with the $\vec{c}$ direction.

Figure 3 shows the evolution in temperature of $\Delta\tau$ for magnetic field oriented along the $\vec{c}$ direction. The quantum oscillation amplitude diminishes rapidly at elevated temperatures, with the signal at 200 mK being an order of magnitude smaller than at 19 mK. Figure 3c

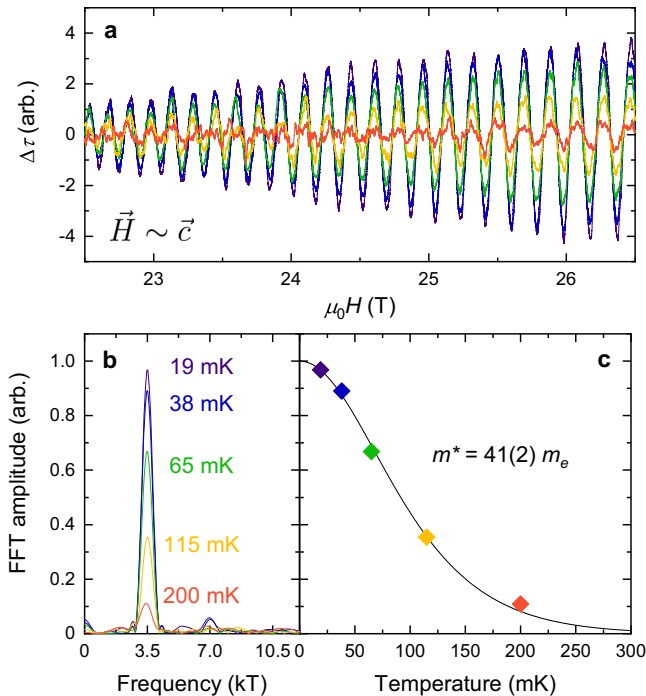

**Fig. 3 | Singular quantum oscillatory frequency and heavy effective cyclotron mass along the $\vec{c}$ direction. a** Oscillatory component of magnetic torque for field applied very close (within 2°) to the $\vec{c}$ direction and **b** corresponding FFT amplitudes for incremental temperatures as indicated. A dominant frequency component of 3.5 kT (and second harmonic at 7.0 kT) is observed. **c** FFT amplitude as a function of temperature (coloured points). The solid line is a fit to the Lifshitz-Kosevich formula[35], which fits the data well and yields a heavy effective cyclotron mass, $m^{*}$, of 41(2) $m_e$, where $m_e$ is the bare electron mass.

fits the quantum oscillatory amplitude to the temperature dependence of the Lifshitz-Kosevich formula[35] (see Methods for details), yielding a heavy effective cyclotron mass, $m^{*}$, of 41(2) $m_e$, consistent with observations reported in ref. 33. For an inclined angle of the magnetic field we observe heavier effective masses up to 78(2) $m_e$ (see Supplementary Information).

We plot the angular evolution of quantum oscillatory frequency with magnetic field tilt angle in Fig. 4 for both the $\vec{c}$ to $\vec{a}$ and $\vec{c}$ to $\vec{b}$ rotation planes. Due to experimental limitations, our $\vec{c}$ to $\vec{b}$ measurements were constrained to within 45° of rotation. We also plot the dHvA frequency simulation for our calculated Fermi surface sections (see Supplementary Information for simulation details), and find remarkably good correspondence between measurement and theory for all frequency branches. Thus, we find that the dHvA profile of UTe$_2$ is excellently described by two quasi-2D Fermi sheets, of 'squircular' (super-elliptical) cross-section, with the hole- (electron-) type sheet centred at the **X** (**Y**) point of the Brillouin zone (Fig. 4c).

We note that in performing DFT calculations (see Supplementary Information) we were unable to capture the angular profile of the low, spectrally dominant frequency branch in the $\vec{c}$ to $\vec{a}$ rotation plane. This branch initially decreases in frequency as field is titled away from $\vec{c}$, reaching a minimum at around 25°, before then increasing rapidly close to $\vec{a}$. This is in sharp contrast to the expectation for a cylindrical Fermi surface section of circular cross-section with no undulations, for which the corresponding frequencies should increase monotonically under rotation away from the cylindrical axis. This property of the quantum oscillatory spectra thus places strong constraints on the possible Fermi surface geometry, and motivated our computation of data-led Fermi surface simulations (see Supplementary Information for simulation details). These simulations yielded cylindrical Fermi sheets with super-elliptical rather than circular cross-sections, with

significant undulation along their lengths, but with a singular cross-sectional area present at extremal points normal to $\vec{c}$. We find that this simulation excellently describes the evolution of the three frequency branches observed at intermediate angles (Fig. 4a).

We can compare the density of states at the Fermi energy inferred from these quantum oscillation experiments with that determined by measurements of the linear specific heat coefficient. Our specific heat measurements (in ambient magnetic field) give a residual normal state Sommerfeld coefficient $\gamma_N$ = 121(1) mJ mol$^{-1}$ K$^{-2}$, consistent with prior reports[11]. Assuming that the quasiparticles of both the hole- and electron-sheets have $m^{*}$ = 41(2) $m_e$ (Fig. 3c), we numerically calculated the contribution to the density of states from both Fermi surface sections (see Methods for calculation). We find that our simulated electron-type sheet should contribute $\approx$61.8 mJ mol$^{-1}$ K$^{-2}$, while the hole-type section should give $\approx$ 58.7 mJ mol$^{-1}$ K$^{-2}$. Together the two cylinders thus comprise a total density of states at the Fermi energy corresponding to a Sommerfeld ratio of $\approx$120.5 mJ mol$^{-1}$ K$^{-2}$, in excellent agreement with the heat capacity measurements of $\gamma_N$. We therefore conclude that these two sheets are very likely the only Fermi surface sections present in UTe$_2$.

A number of studies have sought to reconcile anomalous aspects of the superconducting and normal state properties of UTe$_2$ by evoking models that assume the presence of one or more 3D Fermi surface sections[22–30]. The distinction between having a 3D or quasi-2D Fermi surface is important, as this sets strong constraints on the possible irreducible representations of the point group symmetry for the superconducting order parameter[34]. In the absence of direct observations clarifying the Fermi surface dimensionality these models appeared promising due to their apparent ability to explain physical properties, such as the slight anisotropy of the electrical conductivity tensor[27]. However, these studies did not consider the potential effects of pronounced undulations along the $\vec{c}$ direction of quasi-2D cylindrical Fermi surface sheets. In the Supplementary Information we model the expected components of the electrical conductivity tensor for our simulated Fermi surface[44,45]. Due to the pronounced undulations giving a sizeable component in the $\vec{c}$ direction to the unit vectors normal to the Fermi surface, we find that the low anisotropy of the electrical conductivity of UTe$_2$ can be naturally explained by this pronounced warping of the quasi-2D cylindrical Fermi surface sheets (Fig. 5), without requiring any 3D Fermi surface pocket(s).

Furthermore, our angle-dependent dHvA measurements and corresponding Fermi surface simulation clearly resolve that the Fermi surface of UTe$_2$ comprises two cylindrical sections, possessing quasiparticle effective masses that fully account for the linear specific heat coefficient. Therefore, any potential 3D Fermi surface sections must be very small in size such that their quantum oscillatory frequencies would be very low, or have very high effective masses to necessitate measurement temperatures considerably lower than 19 mK, or possess very high curvature around their entire surface so as to minimise the phase coherence of intersections with successive Landau tubes. However, as the contribution to the density of states per Fermi surface section is directly proportional to the effective mass of the quasiparticles hosted by that section (see Methods), it therefore seems unlikely that UTe$_2$ possesses any 3D Fermi pockets with markedly heavier effective masses than the cylindrical sections we observe in our temperature-dependent measurements.

We add further confidence to our interpretation of the dHvA data by a quasi-2D Fermi surface model by fitting the 0° 19 mK $\Delta\tau$ curve from Fig. 3 as the sum of two distinct oscillatory components representing the hole- and electron-type sections, respectively (see Supplementary Information). The fit yields two sinusoids of identical frequencies (within error). Notably, there is only a very slight phase-smearing contribution, indicative of negligible small-scale corrugations along the lengths of the cylinders. Performing a Fourier analysis of the residual curve obtained by subtracting the fit from the data

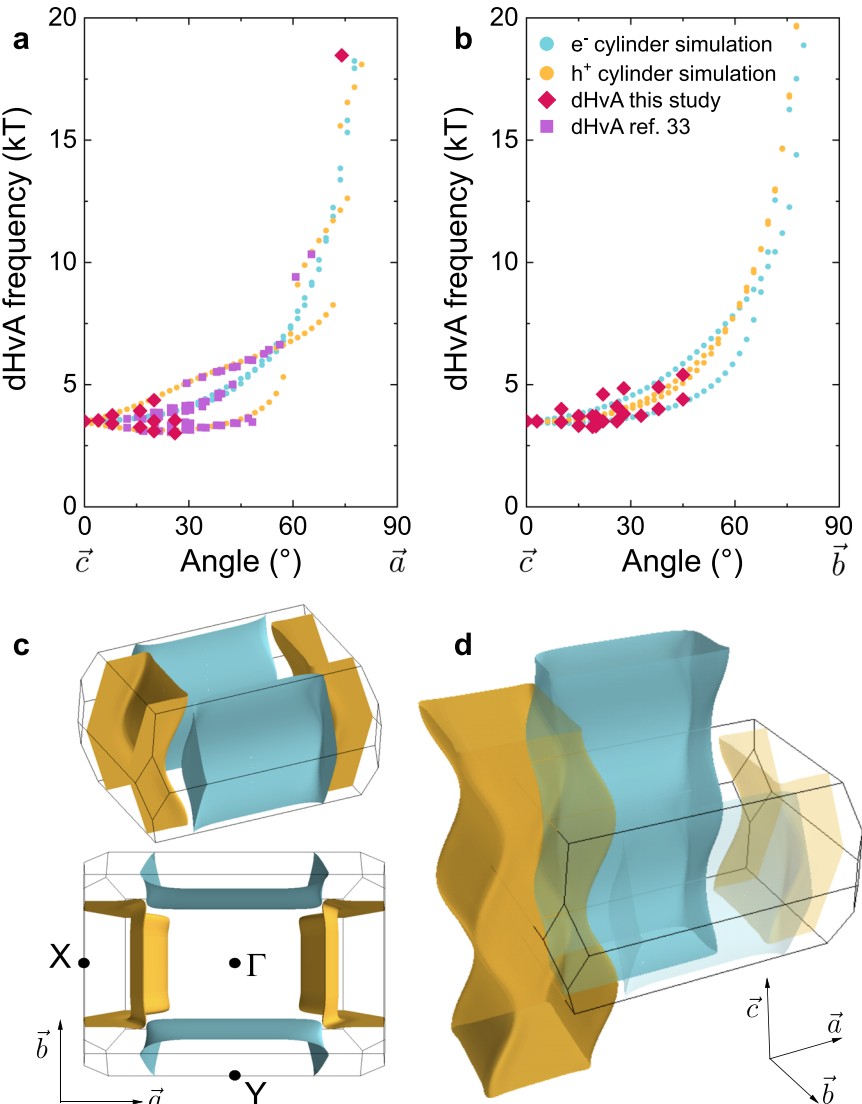

**Fig. 4 | The Fermi surface of UTe₂.** Angular dependence of the dHvA frequencies for **a** the $\vec{c}$ to $\vec{a}$ rotation plane, and **b** the $\vec{c}$ to $\vec{b}$ rotation plane. Blue and gold symbols are simulated frequencies (see Supplementary Information) for extremal orbit areas of our calculated electron-type (e⁻) and hole-type (h⁺) Fermi surface sections, respectively; red and purple symbols represent dHvA data from this study and ref. 33. **c** Side- and top-view of our simulated Fermi surface cylinders, with high symmetry points indicated. Again, blue (gold) represents electron- (hole-) type sections. **d** Extended-zone view of the UTe₂ Fermi surface.

(Supplementary Fig. S15) reveals no additional frequency components that may have been obscured by the large peak in the 0° FFT in Fig. 2b. Therefore, we conclude that our dHvA and SdH data are very well interpreted based on UTe₂ possessing a heavy, quasi-2D, charge-compensated Fermi surface.

## Discussion

Our finding of a quasi-2D Fermi surface in UTe₂ has important implications for determining the symmetry of the superconducting gap structure. Evidence indicating the presence of point nodes has been reported from thermal conductivity measurements[46], with subsequent NMR[47] and scanning SQUID[48] studies interpreting the gap structure within the $D_{2h}$ point group as being of $B_{3u}$ character, with point nodes along the $\vec{a}$ direction. However, recent NMR[40] and thermal conductivity[42] measurements performed on high-quality MSF crystals argue strongly in favour of a highly anisotropic full gap, of the $A_u$ representation. The quasi-2D cylinders of our Fermi surface model would be consistent with either a nodal $B_{3u}$ or fully gapped $A_u$ scenario. Further experimental and theoretical studies to distinguish between these symmetries – or the possibility of a nonunitary combination of

both[16,47,49,50] – are urgently called for to provide a complete microscopic understanding of the superconducting order parameter in UTe₂.

It is interesting to consider how different the Fermi surface of UTe₂ is from the complex multisheet Fermi surfaces found in the ferromagnetic superconductors URhGe, UGe₂, and UCoGe, the latter two of which have small 3D pockets[51]. Contrastingly, the possession of a relatively simple quasi-2D Fermi surface comprising charge-compensated cylindrical components is remarkably similar to other unconventional superconductors, including the Fe-pnictides[52,53], underdoped high-$T_c$ cuprates[54,55], and Pu-based superconductors[56]. It has been suggested[19] that the near nesting between quasi-2D Fermi surface sections favours spin fluctuations in UTe₂ and may thereby strengthen the spin-triplet pairing mechanism. Thus, given the pronounced undulation but negligible degree of small-scale corrugation of the UTe₂ Fermi sheets, this is likely the reason why UTe₂ exhibits such a markedly higher $T_c$ than its ferromagnetic U-based cousins. Given the multitude of theoretical study into the effects of d-wave pairing symmetry hosted by such a Fermi surface in the case of e.g. the cuprates[57], it is therefore interesting to consider what similarities and

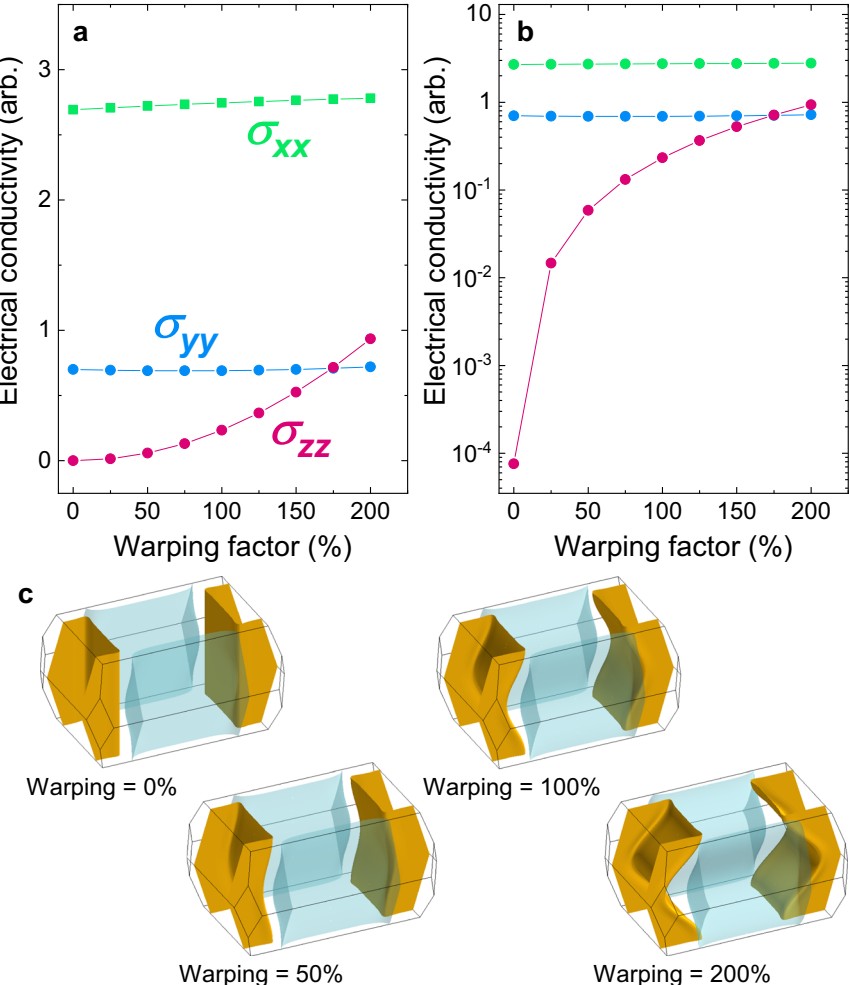

**Fig. 5 | Dependency of electrical conductivity on the warping of the Fermi surface cylinders. a** Diagonal components of the electrical conductivity tensor determined from the Fermi surface fitting (see Supplementary Information for calculation details). $\sigma_{xx,yy,zz}$ respectively refer to conductivity in the $\vec{a}$, $\vec{b}$, $\vec{c}$ directions. 100% warping corresponds to our Fermi surface model of UTe$_2$, which we find best fits the dHvA data (Fig. 4). **b** The same conductivity components as panel **a** plotted here on a logarithmic scale. For 0% warping (no undulation) we would expect $\sigma_{zz}$ to be orders of magnitude lower than $\sigma_{xx}$ and $\sigma_{yy}$. However, for our Fermi surface model (at 100% warping) each of $\sigma_{xx}$, $\sigma_{yy}$, and $\sigma_{zz}$ are within an order of magnitude. **c**, Fermi surface simulations for warping factors between 0 and 200%.

differences may be found when considering instead a *p*-wave symmetry.

At the first-order metamagnetic transition obtained at $\mu_0 H \approx 35$ T for $\vec{H} \parallel \vec{b}$, a Fermi surface reconstruction has been proposed to occur[11] due to the observation of a change of sign of the Seebeck coefficient and a reported discontinuity in the carrier density as interpreted from Hall effect measurements[58]. This raises the interesting possibility that the high field re-entrant superconducting phase[12], which is acutely angle-dependent and appears to persist to at least 70 T (ref. [59]), may be markedly different in character compared to the superconductivity found below 35 T. We note that our Fermi surface simulations predict the occurrence of a Yamaji angle[60] at the magnetic field orientation where the high field re-entrant superconducting phase is most pronounced (see Supplementary Fig. S22), similar to prior reports[59,61]. This implies that a sharp peak in the field-dependent density of states may underpin the microscopic mechanism driving this exotic superconducting state. A further quantum oscillation study beyond the scope of this work, in the experimentally challenging temperature-field regime of $T < 100$ mK and $\mu_0 H > 35$ T, would therefore be of great interest in comparing the underlying fermiology of the magnetic field re-entrant superconducting phase with that of the lower field Fermi surface we uncover here.

We note that while our manuscript was undergoing peer-review, we became aware of a study of the oscillatory magnetoconductance of UTe$_2$ in which the authors interpreted their measurements as being indicative of the presence of small, light, 3D Fermi surface sections[62]. This was surprising, as we observed no signatures of such pockets in our dHvA or SdH measurements, nor did an independent group in their detailed and careful dHvA measurements[33,63]. However, we have since been able to reproduce the magnetoconductance oscillations reported by ref. [62]. We found that such observations could not actually be reconciled with the presence of 3D Fermi sections, but instead can be well understood as a result of high magnetic field-induced quasiparticle tunnelling between the cylindrical sheets, analogous to prior studies of quasi-2D organic metals in which such observations were also present in kinetic properties such as the conductance but, notably, were not observed in derivatives of the free energy such as the magnetization[64,65]. These findings will be reported in detail elsewhere[66].

In conclusion, our quantum oscillation study on pristine quality crystals has revealed the quasi-2D nature of the Fermi surface in UTe$_2$. Performing dHvA and SdH measurements in magnetic field strengths well above the $\vec{c}$-axis upper critical field have enabled us to compute the Fermi surface geometry, which consists of two cylindrical sheets of

super-elliptical cross-section with a significant degree of undulation along their lengths. We find that this pronounced undulation can account for the relatively modest anisotropy of the electrical conductivity tensor, which had previously been cited as evidence favouring the presence of 3D Fermi surface pockets – however, we find no evidence for the presence of any 3D sections. We numerically calculated the contribution to the density of states for our computed Fermi surface, and find that it fully accounts for the normal state Sommerfeld ratio determined from specific heat measurements. Our findings indicate that despite having pronounced axial undulations the Fermi surface of $UTe_2$ possesses a negligible degree of small-scale corrugation, implying that the Fermi sheets may nest very closely together, thereby favouring magnetic fluctuations that enhance the spin-triplet superconducting pairing mechanism.

## Methods

### Sample preparation

High-quality single crystals of $UTe_2$ were grown using the molten salt flux technique adapted from ref. [36]. We used an equimolar mixture of powdered NaCl (99.99%) and KCl (99.999%) salts as a flux, which had been dried at 200 °C for 24 hours. Natural uranium metal with an initial purity of 99.9% was further refined using the solid-state electrotransport (SSE) method[67] under ultra-high vacuum ( ~ $10^{-10}$ mbar); by passing a high electrical current of 400 A through the initial uranium metal, impurities can be removed extremely effectively.

Following SSE treatment, a piece of purified uranium of typical mass ≈0.35 g was etched using nitric acid to remove surface oxides. It was subsequently placed in a carbon crucible of inner diameter 13 mm together with pieces of tellurium (99.9999%) with the molar ratio of 1:1.71; subsequently, the equimolar mixture of NaCl and KCl was added. The molar ratio of uranium to NaCl,KCl mixture was 1:60. The whole process was performed under a protective argon atmosphere in a glovebox. The carbon crucible was plugged by quartz wool, placed in a quartz tube, and heated up to 200 °C under dynamic high vacuum (~$10^{-6}$ mbar) for 12 hours. Then it was sealed and placed in a furnace. It was initially heated to 450 °C in 24 hours, and left there for a further 24 hours. Then it was heated to 950°C at a rate of 0.35°C/min and kept there for an additional 24 hours. Afterwards the temperature was slowly decreased at a rate of 0.03°C/min down to 650°C, maintained there for 24 hours, and then cooled down to room temperature during the following 24 hours.

After the growth process, the ampoules were crushed and the contents of the carbon crucibles were immersed in water where the salts rapidly dissolved. Bar-shaped crystals were manually removed from the solution, rinsed with acetone, and stored under an argon atmosphere prior to characterisation and quantum oscillation studies. The longest edge of the produced single crystals was typically 3-12 mm (along the $\vec{a}$ direction), with widths 0.5-1.2 mm (along $\vec{b}$) and thicknesses around 0.2-1 mm (along $\vec{c}$).

### Capacitive torque magnetometry measurements

Torque magnetometry measurements were performed at the National High Magnetic Field Laboratory, Tallahassee, Florida, USA. Measurements were taken in SCM4 fitted with a dilution refrigerator sample environment. Single crystal samples were oriented with a Laue X-ray diffractometer. We note that our angular data obtained in the $\vec{c}-\vec{a}$ plane is calibrated to within ≈ 2° of experimental uncertainty; however, a possible azimuthal offset in the $\vec{c}-\vec{b}$ angles means that these data should only be taken to be accurate to within ≈5°. Samples were mounted on beryllium copper cantilevers suspended above a copper base plate, thereby forming a capacitive circuit component. The capacitance of the cantilever–base plate system was measured as a function of applied magnetic field strength by a General Radio analogue capacitance bridge in conjunction with a phase-sensitive detector. This configuration of cantilever and base plate was mounted on a

custom-built rotatable housing unit, allowing for the angular dependence of the dHvA effect to be studied.

In analysing the measured torque data, the oscillatory component was isolated from the background magnetic torque by subtracting a smooth monotonic polynomial fit by the local regression technique[68]. The main benefit of this technique over simply subtracting a polynomial fitted over the whole field range is that the LOESS window over which the averaging occurs can be modified; for oscillations of faster (slower) frequency, smaller (larger) LOESS windows will achieve a better isolation of the dHvA effect signal. This averaging window then slides along the entire curve to extract the oscillatory component from the background magnetic torque.

### Lifshitz-Kosevich temperature study

Figure 3 a shows quantum oscillations measured at various temperatures, with the quantum oscillatory amplitude being strongly diminished at elevated temperature. We extract an effective cyclotron mass, $m^*$, by fitting the temperature dependence of the FFT amplitudes to the Lifshitz-Kosevich formula for temperature damping; this fit is plotted in Fig. 3c.

The temperature damping coefficient, $R_T$, may be written as[35]:

$$R_T = \frac{X}{\sinh X} \tag{1}$$

where

$$X = \frac{2\pi^2 k_B T m^*}{e\hbar B}, \tag{2}$$

in which $e$ is the elementary charge, $\hbar$ is the reduced Planck constant, $k_B$ is the Boltzmann constant, $T$ is the temperature, and $B$ is the average magnetic field strength of the inverse field range used to compute the FFTs. Thus $m^*$ can be found by fitting the quantum oscillatory amplitude to Eqn. (1) as a function of temperature.

### Evaluation of the density of states at the Fermi level

The quasiparticle density of states at the Fermi level, $g(E_F)$, may be expressed[69] in terms of the linear specific heat coefficient extrapolated from the normal state, $\gamma_N$, as

$$g(E_F) = \frac{3\gamma_N}{\pi^2 k_B^2}. \tag{3}$$

We can compare this with the density of states predicted for a given Fermi surface geometry as measured by the dHvA effect[35]. For a Fermi surface section with surface element $d\mathcal{S}$, which hosts quasiparticles of effective mass $m^*$ that have Fermi velocity $v_F$, we can write

$$g(E_F) = \frac{1}{4\pi^3\hbar}\int \frac{d\mathcal{S}}{v_F}. \tag{4}$$

For the simple geometric case of a cylindrical Fermi surface, of radius $k_F = \sqrt{k_x^2 + k_y^2}$ and height $k_z$, combining these two expressions gives a contribution (per cylinder) to the linear specific heat coefficient of

$$\gamma_N = \frac{k_B^2 V m^* k_z}{6\hbar^2}, \tag{5}$$

for a metal of molar volume $V$. Therefore, comparing this simple case with our dHvA data, for the 3.5 kT quantum oscillatory frequency observed for magnetic field applied along the $\vec{c}$ direction (Figs. 2, 3), assuming that both cylinders have the same (single) effective mass we

found in our Lifshitz-Kosevich temperature study (Fig. 3) we can estimate a contribution per (circular) cylinder to $\gamma_N$ of $\approx 51.5$ mJ mol$^{-1}$ K$^{-2}$.

Taking this result for the simple case of ideal, circularly cross-sectional cylindrical Fermi surface sections with no $k_z$ warping, we then numerically calculated the actual surface area of the squircular, warped Fermi sheets we generated in our Fermi surface simulations (Fig. 4). We found that the electron-type section possesses a surface area 1.20 times bigger than the case of the simple circular cross-sectional cylinder of the same cross-sectional area normal to $\vec{c}$ (corresponding to a dHvA frequency of 3.5 kT). For the hole-type sheet, we found that its surface area is bigger by a factor of 1.14. Therefore, we obtained values of $\approx 61.8$ mJ mol$^{-1}$ K$^{-2}$ for the electron sheet and $\approx 58.7$ mJ mol$^{-1}$ K$^{-2}$ for the hole sheet, giving a total contribution to $\gamma_N$ of $\approx 120.5$ mJ mol$^{-1}$ K$^{-2}$.

We note that this treatment is only approximate, as we assume a constant $v_F$ along the entire surface of the Fermi sheets, using the value obtained from our measurement of $m^*$ for magnetic field applied along the $\vec{c}$ direction. At inclined angles of magnetic field tilt angle a range of effective masses is observed, from as low as 32 $m_e$ reported in ref. 33, up to the mass of 78 $m_e$ we find in Supplementary Fig. S19. Therefore, this comparison between the density of states implied by specific heat capacity measurements and inferred from observations of the dHvA effect is only approximate, in the absence of a full determination of the profile of $v_F$ along the entire surface of the Fermi surface sections. Further measurements, to carefully determine the evolution in $m^*$ as a function of rotation angle in the $\vec{c} \cdot \vec{a}$ and $\vec{c} \cdot \vec{b}$ rotation planes, would thus be of great value in minimising the uncertainty of this calculation.

In our calculations we used the cyclotron effective mass measured for magnetic field parallel to the axis of the cylinders, as at this orientation the quantum oscillatory amplitude is largest and thus we are sampling a large proportion of the variation of $v_F$ along the cylinders' surfaces. Notably, a special aspect of our Fermi surface model is that all cross-sectional orbits are extremal for field oriented along $\vec{c}$, which likely explains why the signal amplitude is so much larger here. Given the close correspondence between the values of $\gamma_N$ measured by specific heat experiments and calculated from our dHvA data and Fermi surface simulations, this adds strong confidence to our Fermi surface simulations and interpretation of the dHvA data that these two quasi-2D sections likely comprise the only Fermi surface sheets present in UTe$_2$.

## Data availability
The datasets supporting the findings of this study are available from the University of Cambridge Apollo Repository[70].

## Code availability
The custom code used in this study is available here[71,72].

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

## Acknowledgements

We are grateful to J. Chen, D. Shaffer, D.V. Chichinadze, S.S. Saxena, C.K. de Podesta, P. Coleman, T. Helm, A.B. Shick, and W. Luo for fruitful discussions. We thank T.J. Brumm, T.P. Murphy, A.F. Bangura, D.E. Graf, S.T. Hannahs, S.W. Tozer, E.S. Choi, and L. Jiao for technical advice and assistance. This project was supported by the EPSRC of the UK (grant no. EP/X011992/1). A portion of this work was performed at the National High Magnetic Field Laboratory, which is supported by National Science Foundation Cooperative Agreement No. DMR-1644779 and the State of Florida. Crystal growth and characterization were performed in MGML (mgml.eu), which is supported within the program of Czech Research Infrastructures (project no. LM2023065). We acknowledge financial support by the Czech Science Foundation (GACR), project No. 22-22322S. The EMFL supported dual-access to facilities at MGML, Charles University, Prague, under the European Union's Horizon 2020 research and innovation programme through the ISABEL project (No. 871106). A part of this work was also supported by the JAEA REIMEI Research Program. DFT calculations were performed using resources provided by the Cambridge Service for Data Driven Discovery (CSD3) operated by the University of Cambridge Research Computing Service (https://www.csd3.cam.ac.uk), provided by Dell EMC and Intel using Tier-2 funding from the Engineering and Physical Sciences Research Council (capital grant EP/T022159/1), and DiRAC funding from the Science and Technology Facilities Council (https://www.dirac.ac.uk). This project also used

the ARCHER2 UK National Supercomputing Service (https://www.archer2.ac.uk). A.G.E. acknowledges support from a QuantEmX grant from ICAM and the Gordon and Betty Moore Foundation through Grant GBMF5305; from the Henry Royce Institute for Advanced Materials through the Equipment Access Scheme enabling access to the Advanced Materials Characterisation Suite at Cambridge, grant numbers EP/P024947/1, EP/M000524/1 & EP/R00661X/1; and from Sidney Sussex College (University of Cambridge). T.I.W. and A.J.H. acknowledge support from EPSRC studentships (EP/R513180/1 & EP/M506485/1).

## Author contributions

A.G.E., N.J.M.P., A.J.H., R.N. & S.M.B. performed quantum oscillation measurements. P.O., H.S., Y.H. & M.V. optimised the molten salt flux growth technique of $UTe_2$. High-quality single crystal specimens were grown and characterised by A.C., Jiří P., Jan P., T.H., G.B., V.S. & M.V. Quantum oscillation data were analysed by A.G.E., N.J.M.P. & A.J.H., and subsequently interpreted by A.G.E., T.I.W., N.J.M.P., Z.W., A.J.H., G.G.L. & F.M.G. Fermi surface modelling and DFT calculations were performed by T.I.W., who also modelled the electrical conductivity tensor with guidance from G.G.L. The manuscript was written by A.G.E. & T.I.W., with input from all co-authors. A.G.E. conceived and oversaw the project.

## Competing interests

The authors declare no competing interests.
