## [Peer Review File · Nature Communications]

Quasi-2D Fermi surface in the anomalous superconductor UTe₂REVIEWER COMMENTS

Reviewer #1 (Remarks to the Author):

Eaton et al., report dHvA measurements on UTe₂, which is a candidate of a triplet superconductor and is currently investigated very extensively worldwide.

The authors measured dHvA oscillations for various field directions and show that the angular variation of the observed dHvA frequencies can be explained by assuming two Q2D Fermi pockets (cylinders). They also determined the effective masses for some of the frequencies. Using the effective mass for $B // c$, they estimated the Sommerfeld coefficient of the specific heat associated with the two Fermi cylinders, found that the estimate is in good agreement with values estimated in specific-heat measurements, and suggest that the Fermi surface in UTe₂ is most likely composed of only the two observed FS cylinders.

The paper is written well. The used methods are described sufficiently, the presented data are nice, and basic analyses are correct.

The presented data are very consistent with a previous report (ref. 32), and the conclusion of the two Q2D FS cylinders is also consistent with ref. 32.

Although the present authors used higher magnetic fields than ref. 32 and hence could measure dHvA oscillations for field directions close to $B // c$, it did not alter the conclusion of the two FS cylinders.

The present authors make a novel claim that the field angle where the field-reentrant SC occurs may be related to a Yamaji angle where the maximum and minimum cross-sections of the hole cylinder coincides (Extended Data Fig. 6). However, for the c - b rotation (Fig. 4b), how the observed frequencies are related to the maximum or minimum frequencies of the hole cylinder is unclear, and hence the presented angular dependences of the two frequencies are too speculative to make such a claim. In addition, Fig. 3a of ref.51 indicates a rather wide range (~ 45 to 62 deg. of Extended Data Fig. 6) for the reentrant SC, which does not seem to support the authors' claim.

A few specific points.

Fig. 2a The 0-degree oscillation gets smaller above $\sim 26.5T$. Can the authors explain this ?

Fig. 4a Why did you observe no frequency in a field range between ~ 30 and ~ 80 deg., where ref.32 observed many ?

Extended Data Fig. 1

Why don't you fit the 0-deg. data of Fig. 2a? It has a longer field range and the decreasing amplitudes at high fields is suggestive of beating.

I am not convinced that the frequency difference ΔF between the maximum and minimum orbits can be estimated from the fit shown in Extended Data Fig. 1.

Do you really need Eq. 6 to explain the data in Extended Data Fig. 1a ?

Doesn't a single-frequency fit suffice ? Also, you can add a second harmonic.

The FT shows a second harmonic peak, but Eq. 6 neglects the second harmonic.

Please show that the use of Eq. 6 with the Bessel fn. is necessary.

Extended Data Fig. 2

The two effective masses are fairly different. Which is the mass of the hole or electron ?

Despite these differing masses, why do you assume that the effective masses of the electron and hole are the same when you estimate the Sommerfeld coefficient.

Reviewer #2 (Remarks to the Author):

UTe2 Quantum Oscillations – Referee Report

The authors report quantum oscillation studies on UTe₂, a candidate spin-triplet superconductor that attracted enormous attention due to several intriguing properties. The manuscript addresses a question regarding the Fermiology of UTe₂, which is crucial in understanding the details of the pairing mechanism as well as the order parameter symmetry. Here the authors use angle dependent dHvA oscillations measurements on high quality single crystals of UTe₂ to map out the Fermi surface. Using simulations motivated from experimental data and comparison to the density of states obtained from heat capacity data, the authors argue that UTe₂ has two nearly identical quasi-2D Fermi sheets with super-elliptical cross-section with electron- and hole-character each. Importantly, the authors rule out any proposed additional Fermi pockets which are important in understanding the properties of UTe₂ in earlier reports.

The manuscript is technically sound, and the presented analysis is reasonable. The main experimental contribution over previously published work, however, is the addition of the angular dependence of the oscillations when rotating the field from the c-axis to the b-axis and a single data point for field applied 74 degrees from the a-axis rotated toward the b-axis. Although the authors argue that they have found all the features of the FS in UTe₂, it is difficult to prove that there is no 3D pocket that has yet to be observed. Additionally, while they propose a model that fits nicely with the measured angle dependence, it is not a unique solution nor tied to a particular theoretical model. Thus, while useful to the UTe₂ community, it is not clear whether the additional data points and FS modeling are highly relevant to the broader community. The manuscript may be more suitable to publication in a more specialized journal. There are several concerns that need to be addressed before its publication.

1. The authors rule out any additional Fermi pockets based on the comparison of the density of states obtained from the simulation and the heat capacity data. As the authors mentioned in the method session, the estimation of the DOS from the simulated Fermi surface assumes constant Fermi velocity along the tube. Such an assumption is questionable given that a large degree of warping of the Fermi tubes is possible in the simulated model and the fact that the measured effective mass varies by a factor of two. In addition, the authors restrict the analysis solely to zero-degree data arguing that the presented treatment may not accurately describe the super-elliptical cross-section. It would therefore be helpful if the authors could set stricter bounds on the reliability of the DOS comparison. This is particularly important as recent quantum oscillation studies seem to suggest evidence for additional 3D pockets [arXiv:2303.09050]. The 3D pockets observed in that study would have a small contribution to the normal state gamma ($\sim 5 \text{ mJ mol}^{-1} \text{ K}^{-2}$).
2. In Fig.2b, an additional peak observed around 7 kT is assumed as the second harmonic. Does the Lifshitz-Kosevich analysis of this peak result effective mass twice as that of the fundamental?
3. While the simulated FS fit quite well for the data in the a-c plane, the fit is not great for the data in the c-b plane. There are several data points at angles below 30 degrees that lie above the simulated data points. Also, there seems to be a downward curvature in the experimental data at low angles that is not captured by the simulation. Would fitting these features require warping along a second direction?
4. In line 182 to 185, the authors note that the calculated Fermi surface model may account for several effects previously attributed to 3D Fermi surface components. Since this is an important achievement of the simulated model, I encourage the authors to elaborate on this point and briefly mention the several effects.
5. It would be helpful to further clarify why the proposed 2D FS is most compatible with Au or B3u order parameters. It is unclear why a-axis point nodes would be favored over b-axis point nodes just from the FS model.
6. The authors argue that the two FS sections have identical cross section within experimental

resolution using a fit on data obtained with field directly along the c-axis. In the methods section, the authors state that the uncertainty in angle is 2 degrees. Could a small amount of misalignment cause the two FS to appear to have the same cross section? Is the proposed FS consistent with the hall data that seems to show electron carriers are dominant [Niu et al., Phys. Rev. R 2, 033179 (2020)]? Does the relative amplitude of the two contributions match with the relative amplitude at slightly higher angles where the difference in frequency can be resolved? It seems like a single term would also fit the data reasonably well.

Minor comments:

7. Aside from Sr₂RuO₄ (which no longer looks promising) it would be helpful to mention other triplet or potential f/p-wave superconductors in the intro (e.g., UPt₃).

8. The NMR data in reference 7 does not match any other measurement of NMR in UTe₂ and was likely measuring something extrinsic (knight shift of ~0% compared to at least 4% in other studies depending on direction). It would be preferable not to cite Ref. 7 when discussing NMR in UTe₂.

9. The statements in lines 44-46 "Remarkably, recent inelastic neutron...may both be unique to the material" is vague and could use further clarification. What about the pairing mechanism and superconducting phase are unique?

10. There is a typo on line 56: "dominat" instead of dominant.

11. In Fig. 1a, it is useful to mention the residual heat capacity in the superconducting state to compare the sample quality in addition to RRR. In Fig. 1b, the current direction used for the measurements should be mentioned. In the inset of Fig 1b the sample does not appear to go superconducting even though there are data points below 2 K.

12. Line 85: Authors mention that the CVT grown samples typically have T_c around 1.6 K. This is not true as CVT grown samples show a range of T_c values up to around 2 K depending on the temperature gradient and starting composition [Ref. 34].

13. A potential Yamaji angle has been mentioned in the context of UTe₂ previously. Perhaps those references could also be cited when discussing the potential of a Yamaji angle.

14. Is there some reason that mean free path was not estimated using the Dingle damping term?

Reviewer #3 (Remarks to the Author):

UTe₂ has attracted much attention over the past four years, due, for example, to its probable p-wave superconductivity and remarkable field-induced superconducting phase. Any paper that gives valid clues about the underlying normal state is therefore very welcome and deserves publication in a prominent journal. The current manuscript describes de Haas-van Alphen oscillations in the low-field normal state, and represents a useful attempt to characterize the Fermi surface.

Sadly, the de Haas-van Alphen oscillations are only observed over a restricted range of angle and field (though an improvement compared to Aoki's study), leaving significant ambiguities of interpretation and the possibility of missing low quantum-oscillation frequencies. Nevertheless, the authors make a worthy attempt to constrain the Fermi-surface shape using their data, leading to the wavy cylinders of almost rectangular cross-section shown in Figure 2.

More seriously, torque, the technique used to observe the de Haas-van Alphen oscillations in this work, is notably insensitive to three-dimensional Fermi-surface pockets. The signal is proportional

to MxB , and so if there are quasi-two-dimensional pockets, their signature will completely obscure that due to almost spherical pockets. Recent results reported on arXiv and at the APS March Meeting used conductivity measurements (Shubnikov-de Haas) oscillations and wider field ranges to detect what looks like several spherical-ish pockets with much lighter masses. These (light mass compensates for small size) appealingly explain the rather isotropic conductivity of UTe₂ measured by many authors.

In summary of these criticisms, the paper is worth publishing, but its very over-confident statements about the Fermi surface ONLY consisting of quasi-two-dimensional sections need to be retracted before going to press.

The statement in the abstract about analogies with underdoped cuprates (which only have a single carrier species, see e.g., work by Mun et al., 2016) is misleading. Moreover, the pnictides have a much more warped Fermi surface than the sheets measured here, resulting in significant velocity components along the c axis, which of course leads to the rather isotropic $T=0$ upper critical fields for which they are famous. So these analogies with UTe₂ are not really appropriate without detailed qualification. They should be removed from the abstract before publication.

Reviewer responses

We thank the anonymous reviewers for their careful reading of our manuscript and for their helpful feedback. We attach overleaf a new draft wherein these thoughtful comments have now been addressed. We have also added two new figures to the Supplementary Information (Figs. S9 & S16) showing SdH oscillations in the contactless resistivity, and Fourier analysis of a longer field sweep. Both of these new figures fully support the findings of the original manuscript. We hope that the reviewers will agree with us that having performed these revisions, the content of the article has been enhanced and clarified, and the readability is now much improved.

Reviewer #1 comments:

Eaton et al., report dHvA measurements on UTe₂, which is a candidate of a triplet superconductor and is currently investigated very extensively worldwide.

The authors measured dHvA oscillations for various field directions and show that the angular variation of the observed dHvA frequencies can be explained by assuming two Q2D Fermi pockets (cylinders). They also determined the effective masses for some of the frequencies. Using the effective mass for B // c, they estimated the Sommerfeld coefficient of the specific heat associated with the two Fermi cylinders, found that the estimate is in good agreement with values estimated in specific-heat measurements, and suggest that the Fermi surface in UTe₂ is most likely composed of only the two observed FS cylinders.

The paper is written well. The used methods are described sufficiently, the presented data are nice, and basic analyses are correct.

The presented data are very consistent with a previous report (ref. 32), and the conclusion of the two Q2D FS cylinders is also consistent with ref. 32.

Although the present authors used higher magnetic fields than ref. 32 and hence could measure dHvA oscillations for field directions close to B // c, it did not alter the conclusion of the two FS cylinders.

The present authors make a novel claim that the field angle where the field-reentrant SC occurs may be related to a Yamaji angle where the maximum and minimum cross-sections of the hole cylinder coincides (Extended Data Fig. 6). However, for the c-b rotation (Fig. 4b), how the observed frequencies are related to the maximum or minimum frequencies of the hole cylinder is unclear, and hence the presented angular dependences of the two frequencies are too speculative to make such a claim. In addition, Fig. 3a of ref.51 indicates a rather wide range (~45 to 62 deg. of Extended Data Fig. 6) for the reentrant SC, which does not seem to support the authors' claim.

A: We thank the reviewer for their detailed reading of our manuscript, and for their positive feedback concerning the writing, methods, data, and analysis we present.

Concerning the author's point pertaining to the discussion of a possible Yamaji angle coinciding with reentrant superconductivity, as Reviewer #2 pointed out in their Q13, this idea is not novel and has been discussed previously in the text of T. Helm et al and F. Honda et al. To our knowledge, our manuscript is the first instance where an actual plot of simulated dHvA frequency vs angle is

presented, showing the occurrence of the crossing between three of the four frequency branches around the location where the reentrant superconducting phase is strongest.

The reviewer is correct by noting that the reentrant superconducting phase spans a relatively wide angular range. However, the interesting feature of the proposed Yamaji angle hypothesis is that it occurs where this anomalous phase is strongest (i.e. where it persists to the highest applied magnetic field strength, of 68 T in the case of T. Helm et al). We thank the reviewer for highlighting this, and have now amended the discussion to better reflect this. We hope that the reviewer finds this rephrasing of this discussion much clearer in this respect.

Q1: A few specific points. Fig. 2a The 0-degree oscillation gets smaller above ~ 26.5 T. Can the authors explain this?

A1: We thank the reviewer for their careful inspection of our manuscript. Regarding this small feature of our dataset, we regret that this is very likely a spurious artifact of the dilution fridge's temperature stability. Our experimental procedure for measuring each angle point was to sweep the field down to 1 T, where the Hall sensor can obtain a good reading. The magnet was then swept back up to 28 T at a comparatively fast sweep rate (0.5 T/min), taking almost an hour. An effect of this faster sweep rate is that eddy currents in the mixing chamber raise the temperature from base (~ 0.02 K) up to ~ 0.1 K. After returning to 28 T, we then waited typically another hour for the cooling power of the fridge to dissipate this additional heat load. Unfortunately, in the instance mentioned by the reviewer, some cooling was very likely still occurring over the field interval of approximately 27-28 T. Due to the heavy effective mass, this regrettably leads to a slight extraneous diminution of the oscillatory amplitude. Having unfortunately only realised this after the measurement, in performing our Lifshitz-Kosevich analysis of the temperature-dependent oscillatory amplitude we were very careful to select the field range of 22-26.5 T in Fig 3, to ensure that the sample and mixture (and thermometer) were in thermal equilibrium, to give reliable results.

Q2: Fig. 4a Why did you observe no frequency in a field range between ~ 30 and ~ 80 deg., where ref.32 observed many?

A2: Due to the slow sweep rate of the all-superconducting magnet we used to perform these experiments (each angle point took several hours to measure), and the short time available to experimentalists visiting facilities such as the Maglab, we elected to mainly measure angles not previously investigated by Aoki et al. In particular, we devoted considerable time to measurements very close to the *c* axis, and very far away from it, where the high field of this magnet allowed us to carefully examine the dHvA evolution above H_{c2} . We made sure to overlap a few angles with the prior study of Aoki et al, to ensure that the data reproduced, which it did. We also invested considerable time investigating the *c-b* rotation plane as well.

Q3: Extended Data Fig. 1

Why don't you fit the 0-deg. data of Fig. 2a? It has a longer field range and the decreasing amplitudes at high fields is suggestive of beating.

A3: We thank the reviewer for raising this point. We hope the reviewer finds that our answer A1 above, concerning the experimental limitations of this experiment, adequately answers this point as well.

Q4: I am not convinced that the frequency difference ΔF between the maximum and minimum orbits can be estimated from the fit shown in Extended Data Fig. 1. Do you really need Eq. 6 to explain

the data in Extended Data Fig. 1a ? Doesn't a single-frequency fit suffice? Also, you can add a second harmonic. The FT shows a second harmonic peak, but Eq. 6 neglects the second harmonic. Please show that the use of Eq. 6 with the Bessel fn. is necessary.

A4: We are immensely grateful to the reviewer for their comments regarding this section of our manuscript, because it is clear that we did not present this analysis in a coherent manner. For the avoidance of doubt: this discussion was intended to show that the full analytical description is **not** required to describe the data. As the reviewer correctly states, the difference in frequencies is indeed negligible. The interesting finding is that both frequencies appear to be degenerate (within resolution). Our reasoning for fitting to the fully general form for two cylinders of unconstrained area and corrugation was in order to demonstrate that no difference in frequencies, and no effects of corrugation, are observable in the data. We apologise for the convoluted manner in which we previously phrased this discussion. We have now completely rewritten this section, which has moved to the Supplementary Information.

Q5: Extended Data Fig. 2

The two effective masses are fairly different. Which is the mass of the hole or electron ?

A5: This is a good question. Unfortunately, from dHvA experiments alone, it is not possible to deduce whether a particular frequency emanates from a hole- or electron-type quasiparticle orbit. Previous DFT calculations have attributed a heavier bandmass to the hole sheet. However, we would be hesitant to ascribe that the heavier mass we observe is definitely from hole-type carriers until further calculations, consistent with the dHvA data, have been performed.

Q6: Despite these differing masses, why do you assume that the effective masses of the electron and hole are the same when you estimate the Sommerfeld coefficient.

A6: In short, we chose to perform the calculation using the mass obtained for field applied parallel to the cylindrical axes because here the dHvA amplitude is considerably larger than at other angles (Fig. 2). This is because collinear to the axes, a large number of k_z states are contributing to the dHvA signal, therefore we are sampling a large proportion of the variation of v_F along the cylinders' surfaces. We expected that the two cylinders may exhibit quite different masses for this field orientation, but as we show in Fig 3 a single-component LK fit fits very well, indicating that the masses of both sheets are very similar here. This is consistent with unpublished data reported by D. Aoki at the recent APS March meeting, which showed masses in the range of 30-50 m_e close to the c -axis. Therefore, we are confident that the use of our determined value of 41 m_e is well justified for performing the comparison with the Sommerfeld coefficient. We note that the heavier masses we observe at an inclined tilt angle – at which point the oscillatory amplitude is much smaller as we are only sampling a thin slice of k states – may imply a rather inhomogeneous variation of v_F , perhaps especially so at different max/min of the undulations of the cylinders, which will be interesting to follow up in future studies. However, this inhomogeneity is not so pertinent when comparing to the Sommerfeld coefficient – it is the average of v_F that is of most importance, and thus sampling the maximal number of k states, where the oscillatory signal is by far the largest, is the best way to achieve this. Hence, this was the procedure we employed.

Reviewer #2 comments:

The authors report quantum oscillation studies on UTe₂, a candidate spin-triplet superconductor that attracted enormous attention due to several intriguing properties. The manuscript addresses a question regarding the Fermiology of UTe₂, which is crucial in understanding the details of the pairing mechanism as well as the order parameter symmetry. Here the authors use angle dependent dHvA oscillations measurements on high quality single crystals of UTe₂ to map out the Fermi surface. Using simulations motivated from experimental data and comparison to the density of states obtained from heat capacity data, the authors argue that UTe₂ has two nearly identical quasi-2D Fermi sheets with super-elliptical cross-section with electron- and hole-character each. Importantly, the authors rule out any proposed additional Fermi pockets which are important in understanding the properties of UTe₂ in earlier reports.

The manuscript is technically sound, and the presented analysis is reasonable. The main experimental contribution over previously published work, however, is the addition of the angular dependence of the oscillations when rotating the field from to c-axis to the b-axis and a single data point for field applied 74 degrees from the a-axis rotated toward the b-axis. Although the authors argue that they have found all the features of the FS in UTe₂, it is difficult to prove that there is no 3D pocket that has yet to be observed. Additionally, while they propose a model that fits nicely with the measured angle dependence, it is not a unique solution nor tied to a particular theoretical model. Thus, while useful to the UTe₂ community, it is not clear whether the additional data points and FS modeling are highly relevant to the broader community. The manuscript may be more suitable to publication in a more specialized journal. There are several concerns that need to be addressed before its publication.

A: We thank the reviewer for their careful reading of our manuscript, and for recognising that understanding the Fermiology of UTe₂ is “crucial in understanding the details of the pairing mechanism as well as the order parameter symmetry” of this material. We appreciate that the reviewer recognises the importance of our analysis at the 74 degree point – however, we would argue that our measurements close to the c-axis are probably more important due to what this tells us about the geometry of the two cylinders. We believe our addition in this revision of SdH data also strongly enhances our dataset. We hope that, following our edits to the discussion concerning Fig. S10 taking on board the helpful comments of R1, that R2 will also find this portion of our manuscript to be much improved in this revision.

Q1: The authors rule out any additional Fermi pockets based on the comparison of the density of states obtained from the simulation and the heat capacity data. As the authors mentioned in the method session, the estimation of the DOS from the simulated Fermi surface assumes constant Fermi velocity along the tube. Such an assumption is questionable given that a large degree of warping of the Fermi tubes is possible in the simulated model and the fact that the measured effective mass varies by a factor of two. In addition, the authors restrict the analysis solely to zero-degree data arguing that the presented treatment may not accurately describe the super-elliptical cross-section. It would therefore be helpful if the authors could set stricter bounds on the reliability of the DOS comparison. This is particularly important as recent quantum oscillation studies seem to suggest evidence for additional 3D pockets [arXiv:2303.09050]. The 3D pockets observed in that study would have a small contribution to the normal state gamma (~5 mJ mol⁻¹ K⁻²).

A1: Answer Redacted

Q2: In Fig.2b, an additional peak observed around 7 kT is assumed as the second harmonic. Does the Lifshitz-Kosevich analysis of this peak result effective mass twice as that of the fundamental?

We thank the reviewer for raising this point. Yes, the temperature evolution of this frequency peak evolves as one would expect for the second harmonic. This is something we checked previously, to ensure its attribution as a second harmonic was valid. However, because it is so heavy, only the 19 mK and 38 mK temperature sweeps can clearly resolve it – at 65 mK it descends into the noise floor. Taking the Fourier weight from these three temperatures yields a fitted mass of $(93 \pm 21) m_e$, consistent with being double the mass of $(41 \pm 2) m_e$ we observe for the fundamental. Owing to the large error bar, we did not feel that this measurement of the effective mass of the second harmonic was of sufficient quality to merit its explicit inclusion in the final manuscript.

Q3: While the simulated FS fit quite well for the in the a-c plane, the fit is not great for the data in the c-b plane. There are several data points at angles below 30 degrees that lie above the simulated data points. Also, there seems to be a downward curvature in the experimental data at low angles that is not captured by the simulation. Would fitting these features require warping along a second direction?

A3: We thank the reviewer for highlighting this point. The principal reason why we have been able to discern the geometry of the UTe_2 Fermi surface in detail, and why the prior study by D. Aoki et al was not, is because we measured the dHvA effect right along the *c*-axis, and far away (at 74deg) in the *c-a* plane. This allowed us to tightly constrain the associated geometry of the Fermi sheets for rotations in this plane. Unfortunately, we do not have a similar point at high angle in the *c-b* plane. Therefore, there is some uncertainty in this direction. The reviewer asks whether warping would be required along a second direction, however if one inspects closely, the electron-type cylinder warps one way while the hole-type one warps the other. Thus, while we have very well constrained the warping of the hole cylinder, follow up studies measuring to higher angles in the *c-b* plane would be useful in terms of better constraining the warping of the other cylinder. However, overall on this minor point, we would argue that it is only a very small proportion of points that are not captured, and that our conclusions are in no way affected by this.

Q4: In line 182 to 185, the authors note that the calculated Fermi surface model may account for several effects previously attributed to 3D Fermi surface components. Since this is an important

achievement of the simulated model, I encourage the authors to elaborate on this point and briefly mention the several effects.

A4: We thank the reviewer for raising this point, and for encouraging us to elaborate on the features of our model. We are currently working on a follow up study utilising Eliashberg calculations to better understand the Fermiology of UTe_2 . Our initial computation of the electrical conductivity tensor, using our simulated Fermi surface presented in this manuscript, yields very good correspondence with published studies. We have added some short phrasing, as encouraged by the reviewer, to remark on this feature (that is only possible due to the pronounced undulations of the cylinders, which allows for 3D-like properties to be manifested from a quasi-2D Fermi surface).

Q5: It would be helpful to further clarify why the proposed 2D FS is most compatible with Au or B3u order parameters. It is unclear why a-axis point nodes would be favored over b-axis point nodes just from the FS model.

A5: We are grateful that the reviewer raises this point, as there is clearly some confusion, indicating that our initial phrasing of this discussion was unclear. When we referred to “these observations” on line 200 of our original submission, we were referring to the scanning SQUID, transport, and NMR measurements discussed in the previous sentence. The reviewer is of course correct that the Fermi surface model alone does not have any preference for point nodes along the a - or b -axes: they just cannot be along the c -axis. We have now rewritten this paragraph in our revised manuscript. We hope the reviewer finds our rephrasing of this discussion to be much clearer.

Q6: The authors argue that the two FS sections have identical cross section within experimental resolution using a fit on data obtained with field directly along the c -axis. In the methods section, the authors state that the uncertainty in angle is 2 degrees. Could a small amount of misalignment cause the two FS to appear to have the same cross section? Is the proposed FS consistent with the hall data that seems to show electron carriers are dominant [Niu et al., Phys. Rev. R 2, 033179 (2020)]? Does the relative amplitude of the two contributions match with the relative amplitude at slightly higher angles where the difference in frequency can be resolved? It seems like a single term would also fit the data reasonably well.

A6: We thank the reviewer for this thoughtful consideration of our analysis of the oscillatory waveform. In response to Q4 from Reviewer 1, we have considerably rewritten the discussion preceding Fig. S10 in the revised version of our manuscript. We hope that Reviewer 2 also finds this rephrasing to be a marked improvement on the original version. Regarding the work of Niu et al, when they subtract the anomalous Hall component to isolate the ordinary part of R_H , they find that $-0.1 < R_0 < 0.1 \mu\Omega \text{ cm T}^{-1}$ (Fig 5). This is consistent with our Fermi surface model, in which both cylinders have approximately equal volume. (We note that the recent revised version of the Helm et al preprint has cast considerable doubt on the interpretation of Hall effect signal given in Niu et al.) Regarding the question of misalignment, it would be quite surprising if the frequencies happened to collapse onto one large amplitude frequency component slightly away from the c -axis. Even if this were the case, the cylinders cannot be misaligned with respect to each other – therefore they would still have the same area (albeit at some small angle of inclination away from the crystallographic axis). However, the inclusion of our SdH measurements in Fig. S9 of our revised manuscript makes a misalignment away from c look very unlikely, given the strong correspondence between this measurement and the data presented in Fig 3 for our torque experiment.

Minor comments:

Q7: Aside from Sr₂RuO₄ (which no longer looks promising) it would be helpful to mention other triplet or potential f/p-wave superconductors in the intro (e.g., UPt₃).

A7: We thank the reviewer for highlighting how our introduction could be improved by broadening the discussion to other candidate materials. We have added some phrasing to this effect.

Q8: The NMR data in reference 7 does not match any other measurement of NMR in UTe₂ and was likely measuring something extrinsic (knight shift of ~0% compared to at least 4% in other studies depending on direction). It would be preferable not to cite Ref. 7 when discussing NMR in UTe₂.

A8: We are especially grateful to the reviewer for raising this important point that we had previously overlooked. We have now removed this reference, and will be sure to exercise caution when discussing such reports in the future.

Q9: The statements in lines 44-46 "Remarkably, recent inelastic neutron...may both be unique to the material" is vague and could use further clarification. What about the pairing mechanism and superconducting phase are unique?

A9: We thank the reviewer for identifying this vague passage in our initial manuscript. In the interests of brevity, we have removed it from our revised version.

Q10: There is a typo on line 56: "dominat" instead of dominant.

A10: We thank the reviewer for their close reading of the manuscript and spotting this error, which we have now rectified.

Q11: In Fig. 1a, it is useful to mention the residual heat capacity in the superconducting state to compare the sample quality in addition to RRR. In Fig. 1b, the current direction used for the measurements should be mentioned. In the inset of Fig 1b the sample does not appear to go superconducting even though there are data points below 2 K.

A11: We thank the reviewer for raising these points concerning the discussion of sample quality from characterisation measurements. We now specify the direction of applied current in the caption of Figure 1. Concerning the determination of the residual heat capacity in the superconducting state, while this would be highly desirable to obtain, unfortunately this measurement did not extend to a sufficiently low temperature in order to be able to fit that to a meaningful level of accuracy. Regarding the appearance of the inset not showing a superconducting transition, the data in the main panel are also in the inset. With this new generation of very high quality samples, the residual resistivity is so low that it is not so clear to observe the superconducting transition when zoomed out to the level of the inset.

Q12: Line 85: Authors mention that the CVT grown samples typically have T_c around 1.6 K. This is not true as CVT grown samples show a range of T_c values up to around 2 K depending on the temperature gradient and starting composition [Ref. 34].

A12: We would like to respectfully disagree with the reviewer on this point. A large majority of reports on CVT grown samples report T_c values close to 1.6 K. It is only recently, after considerable effort, that higher quality CVT specimens have become available, which we recognise when citing ref. 34 on lines 86-87 of our original submission. Therefore, we feel it is a fair statement to refer to CVT samples as possessing a "typical" T_c of 1.6 K, as this is reported in an extremely large number of articles. For instance, even relatively recently in T. Helm et al [ref. 51 of our original submission], in their recently

updated preprint (dated April 2023) all measurements are reported on a 1.6 K sample. Therefore, we feel that our wording in this respect is entirely appropriate.

Q13: A potential Yamaji angle has been mentioned in the context of UTe₂ previously. Perhaps those references could also be cited when discussing the potential of a Yamaji angle.

A13: We thank the reviewer for raising this point. To our knowledge, our manuscript is the first instance where an actual plot of simulated dHvA frequency vs angle is presented, showing the occurrence of the crossing between three of the four frequency branches around the location where the reentrant superconducting phase is strongest. As we were not aware of any articles showing a similar plot with which to compare, we therefore did not make any citations in our original manuscript. However, in recent discussions with colleagues, we have become aware that the discussion in the text of Helm et al. and Honda et al. of the possibility of a Yamaji angle – which bizarrely are made in both instances without the presentation of any dHvA frequency simulations, DFT calculations, or Fermi surface drawings – were in fact referring to DFT studies presented at conferences last year (despite neither paper referencing this in their bibliographies). In light of this, we have added these citations to our discussion of a Yamaji angle. We look forwards to these calculations being published, so that we can compare their predictions with those of our model.

Q14: Is there some reason that mean free path was not estimated using the Dingle damping term?

The reviewer raises a good point, as the Dingle term is a standard method for estimating the mean free path. However, to calculate it properly, one needs to know both the effective mass from dHvA and the band mass. However, as we go to great lengths to show in our supplement, DFT calculations of UTe₂ fail to properly capture the electronic structure of this material, so we do not have properly deduced band mass values with which to perform the calculation. Therefore, we elected not to include a Dingle calculation, and instead include other estimates of the mean free path in our supplement, calculated from data we collected that are of high accuracy and precision, yielding estimates for the mean free path that are in good agreement with each other. We feel that this is sufficient, and that there would be no gain from making a crude approximation of the Dingle term from incorrectly deduced band masses.

Reviewer #3 comments:

UTe₂ has attracted much attention over the past four years, due, for example, to its probable p-wave superconductivity and remarkable field-induced superconducting phase. Any paper that gives valid clues about the underlying normal state is therefore very welcome and deserves publication in a prominent journal. The current manuscript describes de Haas-van Alphen oscillations in the low-field normal state, and represents a useful attempt to characterize the Fermi surface.

Sadly, the de Haas-van Alphen oscillations are only observed over a restricted range of angle and field (though an improvement compared to Aoki's study), leaving significant ambiguities of interpretation and the possibility of missing low quantum-oscillation frequencies. Nevertheless, the authors make a worthy attempt to constrain the Fermi-surface shape using their data, leading to the wavy cylinders of almost rectangular cross-section shown in Figure 2.

A: We are grateful to the reviewer for their positive appraisal of our work, and for recognising how our findings help to extend our collective understanding of the normal state Fermiology of UTe₂. Regarding the reviewer's comment pertaining to the limited field range of quantum oscillation measurements presented in our original manuscript, we have taken this point on board. In our revised

version, we have added a new figure (Fig. S16) over an extended field range, to show that no lower frequency oscillations are observed in our experiment. We are grateful to the reviewer for highlighting this aspect in which our manuscript could be improved.

Q1: More seriously, torque, the technique used to observe the de Haas-van Alphen oscillations in this work, is notably insensitive to three-dimensional Fermi-surface pockets. The signal is proportional to MxB , and so if there are quasi-two-dimensional pockets, their signature will completely obscure that due to almost spherical pockets. Recent results reported on arXiv and at the APS March Meeting used conductivity measurements (Shubnikov-de Haas) oscillations and wider field ranges to detect what looks like several spherical-ish pockets with much lighter masses. These (light mass compensates for small size) appealingly explain the rather isotropic conductivity of UTe₂ measured by many authors.

A1: The reviewer raises an excellent point, very similar to the first point voiced by Reviewer #2, concerning SdH measurements attributed to the presence of light spherical-ish pockets. In our answer to Reviewer #2 above (A1) we have gone to great detail to outline our recent measurements, and explain how they are consistent with the quasi-2D Fermi surface reported in this manuscript – and not in fact indicative of any 3D pocket(s). Furthermore, our new manuscript draft contains a new supplementary figure (S9) showing SdH oscillations for field oriented along the c -axis. The SdH response is fully consistent with the dHvA c -axis data presented in the main text. As the reviewer correctly points out, the SdH effect would be sensitive to the presence of any (either exactly or almost) spherical pockets. Given that no additional frequencies are observed in the SdH FFT spectra, this adds very good confidence to our initial interpretation of our dHvA data. We can now therefore conclude, to a very high level of certainty, that the UTe₂ Fermi surface possesses only these two cylindrical sections.

Q2: In summary of these criticisms, the paper is worth publishing, but its very over-confident statements about the Fermi surface ONLY consisting of quasi-two-dimensional sections need to be retracted before going to press.

A2: Following on from our A1 answer above, we hope that the reviewer is convinced by our new SdH oscillations included in Fig. S9, and by our analysis of breakdown orbits by the quantum interference effect given in A1 to Reviewer #2 above, both of which are fully consistent with our quasi-2D Fermi surface simulation. There is simply no evidence from quantum oscillation spectra – measured by us or reported by others – for any 3D Fermi pocket(s). Given these new results, and the excellent correspondence between specific heat and effective mass for our wavy cylinders, we can now conclude quite firmly that the Fermi surface of UTe₂ comprises solely of these two quasi-2D sheets.

Q3: The statement in the abstract about analogies with underdoped cuprates (which only have a single carrier species, see e.g., work by Mun et al., 2016) is misleading. Moreover, the pnictides have a much more warped Fermi surface than the sheets measured here, resulting in significant velocity components along the c axis, which of course leads to the rather isotropic $T=0$ upper critical fields for which they are famous. So these analogies with UTe₂ are not really appropriate without detailed qualification. They should be removed from the abstract before publication.

A3: We thank the reviewer for raising this point, which we agree many readers may have found confusing. So as to not detract from the main focus of our manuscript, we have removed this passage from the abstract, as the reviewer recommended.

REVIEWER COMMENTS

Reviewer #2 (Remarks to the Author):

Through the revision of the paper and in the response letter to the first round of the reports, the authors have done a good job of addressing the concerns raised by the referees. While we still find a very minor point of contention (noted below), it should not delay publication of the manuscript. The central question is whether the revised manuscript is of enough significance outside of the UTe₂ community to warrant publication in Nature Communications.

The main claim in the manuscript is that the Fermi surface of UTe₂ consists of ONLY the quasi-2D sheets the authors detect in the quantum oscillation studies. This is argued based on the comparison of the density of states estimated from the simulations and the heat capacity data. As mentioned in the previous round, such analysis alone cannot rule out the possibility of small 3D-pockets. The quasi 2D sheets in the FS have been previously reported, although the authors present a more detailed model including the undulation along c-axis. Because the experimental detection or exclusion of small 3D-pockets is challenging, an important question is whether the presented FS model can capture relevant properties of UTe₂. The authors state that the Fermi surface cylinders in their model can account for several effects previously attributed to a 3D Fermi surface component, such as the nearly isotropic electrical resistivity, and that an explanation will appear in a later manuscript. However, without such a discussion, the present results do not significantly further the current understanding of the Fermiology of UTe₂ or address open questions in the field.

A large addition to the manuscript over the previous version is the SdH oscillation data from PDO measurements in the Supplemental material. This data indeed questions the claims of observation of 3D pockets recently reported in [arXiv:2303.09050]. The authors' efforts to reproduce the data claiming the 3D-pockets is commendable and their new results and analysis are certainly worth publishing. The authors choose, however, to only show data for H||c and 51 degrees from c to b in the supplemental information. The data for field along a-axis showing low-frequency oscillations is only included in the referee response.

In its current form, the manuscript confirms the existence of the quasi-2D FS in UTe₂ and provides some evidence for the lack of a 3D FS by showing a contradiction between PDO measurements on the present samples compared to those from Broyles et al. The quasi 2D FS sections were already reported from prior dHvA measurements and were largely expected even before salt-flux samples enabled quantum oscillation measurements. The relevance of the paper to the broader community would be improved by including either 1) the calculations showing how the corrugation along the c axis can account for the isotropic resistivity, or 2) their explanation of how the breakdown orbits can account for the low frequency oscillations observed by Broyles et al. This would go further toward resolving some of the outstanding issues in UTe₂ and combined with the current manuscript would warrant publication in Nature Communications. Without this additional data or discussion, we would instead recommend publication in Communications Physics.

Minor Issues

1. Regarding the T_c of CVT grown samples, we still disagree with the authors' argument that the CVT grown samples "typically" have T_c around 1.6 K. It is true that the initial CVT samples have T_c around 1.6 K and were highly studied immediately resulting in many publications. It is now clear that the optimized CVT growth results in T_c around 2 K. Acknowledging that CVT grown UTe₂ samples have T_cs that vary seems relatively straightforward. It is not a completely moot point because some of the discrepancies between reports on CVT-grown samples stem from the fact that many of the properties of CVT-grown UTe₂ are quite sensitive to sample quality. The fact that there are still reports coming out with low-T_c CVT samples does not seem relevant to this point. Higher quality CVT samples were available by mid-2020 [see Cairns et al., J. Phys.: Condens. Matter 32 415602 (2020)] or earlier. It is worth pointing out that with the advent of MSF samples, it may be necessary to revisit many of prior experiments performed on CVT samples.

2. The discrepancy between c-axis data here and in Aoki et al. [JPSJ 92, 065002 (2023)] is still concerning. One possibility is slight misalignment in one of the experiments. For the dHvA measurements here, a small misalignment would not be surprising considering the stated 2-5 degree uncertainty in orientation and the possibility of angular deflection of the cantilever. As stated by the authors, this seems unlikely in the PDO measurements given how the sample was mounted directly on a (001) surface, but the signal to noise ratio in the PDO measurement appears much lower. The signal to noise ratio seems better in Aoki et al., and the data covers a larger field range. But no information is given regarding the uncertainty in sample orientation. Finally, in figure S10 there does seem to be a small amount of beating when comparing the data to the fit. It is unfortunate that the authors do not have additional 0 degree torque data to higher field or with more stable temperature, as it seems a bit surprising that the electron and hole cylinders would have exactly the same area.

Reviewer responses v2

Reviewer #2 comments:

Through the revision of the paper and in the response letter to the first round of the reports, the authors have done a good job of addressing the concerns raised by the referees. While we still find a very minor point of contention (noted below), it should not delay publication of the manuscript. The central question is whether the revised manuscript is of enough significance outside of the UTe₂ community to warrant publication in Nature Communications.

The main claim in the manuscript is that the Fermi surface of UTe₂ consists of ONLY the quasi-2D sheets the authors detect in the quantum oscillation studies. This is argued based on the comparison of the density of states estimated from the simulations and the heat capacity data. As mentioned in the previous round, such analysis alone cannot rule out the possibility of small 3D-pockets. The quasi 2D sheets in the FS have been previously reported, although the authors present a more detailed model including the undulation along c-axis. Because the experimental detection or exclusion of small 3D-pockets is challenging, an important question is whether the presented FS model can capture relevant properties of UTe₂. The authors state that the Fermi surface cylinders in their model can account for several effects previously attributed to a 3D Fermi surface component, such as the nearly isotropic electrical resistivity, and that an explanation will appear in a later manuscript. However, without such a discussion, the present results do not significantly further the current understanding of the Fermiology of UTe₂ or address open questions in the field.

A large addition to the manuscript over the previous version is the SdH oscillation data from PDO measurements in the Supplemental material. This data indeed questions the claims of observation of 3D pockets recently reported in [arXiv:2303.09050]. The authors' efforts to reproduce the data claiming the 3D-pockets is commendable and their new results and analysis are certainly worth publishing. The authors choose, however, to only show data for H || c and 51 degrees from c to b in the supplemental information. The data for field along a-axis showing low-frequency oscillations is only included in the referee response.

In its current form, the manuscript confirms the existence of the quasi-2D FS in UTe₂ and provides some evidence for the lack of a 3D FS by showing a contradiction between PDO measurements on the present samples compared to those from Broyles et al. The quasi 2D FS sections were already reported from prior dHvA measurements and were largely expected even before salt-flux samples enabled quantum oscillation measurements. The relevance of the paper to the broader community would be improved by including either 1) the calculations showing how the corrugation along the c axis can account for the isotropic resistivity, or 2) their explanation of how the breakdown orbits can account for the low frequency oscillations observed by Broyles et al. This would go further toward resolving some of the outstanding issues in UTe₂ and combined with the current manuscript would warrant publication in Nature Communications. Without this additional data or discussion, we would instead recommend publication in Communications Physics.

A: We thank the reviewer for their positive appraisal of our response to the first round of referee comments, and for our "commendable" recent measurements of the contactless resistivity of UTe₂, which as we showed previously can be well interpreted by considering only the quasi-2D Fermi surface components we observed by prior dHvA effect measurements. We strongly considered adding our new dataset of quantum interference oscillations to this manuscript – however, we were concerned

that the very different mechanisms of the dHvA effect (from Landau quantisation directly probing the Fermi surface) and quantum interference oscillations (from kinetic magnetic breakdown trajectories indirectly sampling the Fermi surface) may confuse non-specialist readers and thereby detract from the main message of the manuscript. We have therefore decided to write up these data into a separate self-contained manuscript [arXiv:2307.00568], which fully supports the analysis, results and conclusion of the present manuscript.

Furthermore, we are very grateful to the reviewer for astutely recommending that we highlight the ability of our Fermi surface model to account for the modest anisotropy of the electrical conductivity tensor. As we stated previously, we were planning on including this in a separate later manuscript, but on reflection we agree with the reviewer that it is better placed in the present manuscript, and follows naturally from the formulation and subsequent discussion of our UTe_2 Fermi surface model. Accordingly, we have included a new figure (Fig. 5) and accompanying discussion to the main text, showing how the pronounced undulation of the UTe_2 Fermi surface naturally accounts for the lack of anisotropy in the electrical conductivity tensor. Our revised supplement contains the full details of this calculation. We agree with the reviewer that the inclusion of this calculation goes "...further toward resolving some of the outstanding issues in UTe_2 and combined with the current manuscript would warrant publication in Nature Communications." We thank the reviewer again for their very careful reading of our initial manuscript, and for their insightful comments that have markedly improved its quality.

Minor Issues:

1. Regarding the T_c of CVT grown samples, we still disagree with the authors' argument that the CVT grown samples "typically" have T_c around 1.6 K. It is true that the initial CVT samples have T_c around 1.6 K and were highly studied immediately resulting in many publications. It is now clear that the optimized CVT growth results in T_c around 2 K. Acknowledging that CVT grown UTe_2 samples have T_c s that vary seems relatively straightforward. It is not a completely moot point because some of the discrepancies between reports on CVT-grown samples stem from the fact that many of the properties of CVT-grown UTe_2 are quite sensitive to sample quality. The fact that there are still reports coming out with low- T_c CVT samples does not seem relevant to this point. Higher quality CVT samples were available by mid-2020 [see Cairns et al., J. Phys.: Condens. Matter 32 415602 (2020)] or earlier. It is worth pointing out that with the advent of MSF samples, it may be necessary to revisit many of prior experiments performed on CVT samples.

A2: We thank the reviewer for raising this point again, and for clarifying to us the implications of various sample optimization procedures on the observed physical properties of UTe_2 . We agree with the reviewer that this is an important point to mention, and have therefore altered our phrasing accordingly. We are also grateful to the reviewer for highlighting that "with the advent of MSF samples, it may be necessary to revisit many of prior experiments performed on CVT samples." We have also added some phrasing to this effect in our introduction.

2. The discrepancy between c-axis data here and in Aoki et al. [JPSJ 92, 065002 (2023)] is still concerning. One possibility is slight misalignment in one of the experiments. For the dHvA measurements here, a small misalignment would not be surprising considering the stated 2-5 degree uncertainty in orientation and the possibility of angular deflection of the cantilever. As stated by the authors, this seems unlikely in the PDO measurements given how the sample was mounted directly on a (001) surface, but the signal to noise ratio in the PDO measurement appears much lower. The

signal to noise ratio seems better in Aoki et al., and the data covers a larger field range. But no information is given regarding the uncertainty in sample orientation. Finally, in figure S10 there does seem to be a small amount of beating when comparing the data to the fit. It is unfortunate that the authors do not have additional 0 degree torque data to higher field or with more stable temperature, as it seems a bit surprising that the electron and hole cylinders would have exactly the same area.

A2: We thank the reviewer for raising this point. Regarding the comparison of signal to noise ratio between our SdH measurements and the field modulation measurements of Aoki et al. [JPSJ 92, 065002 (2023)], we find that these are extremely similar when comparing the ratio between the height of the (largest) FFT peak (labelled α_2 in Aoki et al.) with the low frequency noise. Digitising the data of [JPSJ 92, 065002 (2023)] we find this ratio to be 7.15, very similar to the ratio of 7.30 we obtain for our SdH measurements presented in Fig. S9. We note that our torque measurements have considerably better resolution, with the ratio between the large sharp FFT peak and the largest extraneous low frequency noise feature in Fig. S12 being 32.3. However, the reviewer makes a good remark in that some slight deflection of the cantilever away from the expected angle is not impossible – therefore we have added some discussion of this starting from line 118 of our revised supplement. However, any such small deflection is almost certainly within the 2° of uncertainty we quote for this measurement. In any case, some slight misalignment for the cylindrical areas normal to the cylindrical axes would not have a substantive impact on the results and conclusions we draw in this work.

REVIEWER COMMENTS

Reviewer #2 (Remarks to the Author):

The authors have addressed all of our outstanding concerns with the manuscript. The addition of an explanation for the relatively low resistivity anisotropy in UTe₂ due to the warping of the cylinders addresses an ongoing question in the field. We feel that the manuscript is now suitable for publication.

Reviewer #4 (Remarks to the Author):

The authors report high field torque measurements of the de Haas van alphen effect in UTe₂. Such measurements are important in order to determine the shape of the Fermi surface and its degree of renormalisation which are themselves important inputs to models to describe the superconductivity in this material. As the authors state, this material has many interesting properties such as very high field re-entrant superconductivity. Many people are working on this compound, so it is highly topical.

In terms of novelty, the data reported here do add to those reported previously by Aoki et al [JPSJ 2022 (Ref 33)]; adding a c->b scan. The angle range where oscillations are detected however, is generally much less than in ref 33 but the current authors do have one single data point at high angle 74 deg, and some at very low angle (<10 deg) which possibly help to constrain their model. This is odd as the present authors state the RRR of their sample is 900 versus 200 in Ref 33 and furthermore their maximum field is much higher, so you would expect large QO and a larger angle range over which they can be observed. The authors do not comment on this (they should).

Presumably, it means there is some other damping mechanism at play, but I cannot speculate as to what it might be. Eaton et al also report an effective mass for the overlapping orbits at zero degrees, whereas Aoki et al report orbit specific masses at a higher angle where the different signals can be separated.

The main advance in the submitted m/s is that the authors have suggested a simple model which agrees with the measured angle dependence of the QO, whereas DFT calculations (here and in Ref 33) do not in detail. This has value but given that it is an empirical model (not based on ab-initio calculations) there are some questions about the uniqueness of the solution.

Specific issues with the paper are

1) In the abstract and elsewhere the authors refer to 'considerable undulation but negligible small-scale corrugation'. It is not clear to me what this means. This issue is made worse by incomplete details about the empirical model used, and issues with the DFT calculations.

Let me comment on the DFT first. The calculated Fermi surface for the various U values in the SI do not appear to have been calculated with sufficient k-points, and/or an inappropriate interpolation scheme. For example, the blue surface in fig S1 has strange pockets and undulations. This is almost certainly caused by insufficient dense k-mesh. This is further evident for the larger value of U, e.g., S3. The now complete surfaces have 'ripples'. This is a well-known problem which results from using linear cartesian interpolation with a bct k-mesh. The points are not equally spaced in cartesian coordinates and so the interpolation function oscillates. It can be solved, either by reevaluating the energies on an equally spaced cartesian mesh, or using an interpolation scheme which respects the bct symmetry. Increasing the density of the mesh is less effective but does help. Note ref 33 does not have these problems.

For the calculations themselves, the authors should state the structural parameters, including the internal positions (z values), and whether these were relaxed or not. The size of the k-mesh used should also be given in the IBZ (probably a factor 8 lower).

When describing the empirical model, the authors should give further details. They state, 'dominant super-ellipse contribution comes from $n = 5$ ', but what values were used? Presumably a

mix of different n values. Were these got from a fit to the DFT? Details should be given.

In general, the DFT actually does a reasonably good job of describing the FS and reproducing the data. However, as the authors note, it does not describe the lower frequency branch of the hole pocket, presumably because the warping of this pocket is less in the DFT than reality. The relation between the empirical model and the DFT needs to be described in more detail. Some suggestions to do this would be (a) show the fitting of the model to the DFT to give the values of n , (b) show how the FS area varies as a function of k_z for the model and the DFT; perhaps pointing out any 'undulations' in the DFT (if they exist) (c) Fit the DFT to the model to show the relationship between the warping parameters. (d) show the positions of the extremal orbits (as a contour line) on the surfaces, so it is clear where the DFT is coming up short.

In reality the model is not simply based on the data, but is constrained by the DFT. They should make this clear. Furthermore, a significant part of the fit constraints do not come from data in the present study, but rather Ref 33.

2) Are there any small pockets? Probably not, but the authors have overstated the accuracy to which they can determine this. First of all they assume both surface have the same mass, which does not vary with k_z . In Ref 33 the authors show that the effective masses of the different orbits differ quite a bit, e.g., at 29 degrees 57 versus 33 (difference is 70%). It could be that this just reflects the change in area, but in any case they authors need to quantify the uncertainty from their approximation. Can they quantify any difference in mass by making a 2 mass fit to R_T ? Secondly, their calculation of γ from their own data is possibly suspect. They used the 2D expression for the relationship between γ and the mass, and then scaled this according to the area of their warped cylinder. It is not obvious to me that that would work (but it might). A better way would be to set v_F for their cylinder to match the measured mass, using $m^* \int dA/dE$, then perform the integration, across the surface.

It is unlikely that after they have properly determined the error in γ , they would be room for a small (say 20-30mJ) contribution from some small 3D pockets. So I don't think they can rule these out.

3) Zero degrees. The authors state that, for example, the mass measurements are taken at zero degrees. The torque is always zero at this point, so this cannot be. In several places they quote the angle accuracy to be 2 deg, but they say there could be a 5 deg misalignment. They should be consistent. Relative error and systematic error. Did the authors measure both positive and negative angles to show any offset? In the methods they should describe the torque factor ($dF/d\theta$). It would be useful (in the SI) to show amplitude as a function of angle.

4) One useful part of the current work is the fact they found no additional orbits. Interestingly they do find an orbit at very high angle (74 deg). However, they do not state what angle range and spacing they searched (e.g. full 90 deg with 5 deg spacing?). I suspect that the one point they find at very high angle is in fact the so called Yamaji point, where the amplitude is massively enhanced. This would throw into question their model and the correspondence between this point and the re-entrant superconductivity. Is this a singular point where QO were observed? This needs to be discussed. Furthermore, they state that 'This implies that a sharp peak in the density of states may underpin the microscopic mechanism driving this exotic superconducting state.'. The density of states is obviously not angle dependent, so it is unclear what they mean by this. The Yamaji point is actually where A_k does not vary with k_{\parallel} , or there is a peak in the so-called curvature factor.

5) Determination of the mean ℓ -free path. The authors have attempted to determine the mean free path by equating the cyclotron radius with the mean free path. This is equivalent to setting $\omega_c \tau = 1$ at this field, so $R_D = \exp(-\pi)$. Although this is called 'approximate' there is no discussion about how 'approximate'. It is further stated that this is a 'lower bound'. Although I agree this gives an order of magnitude estimate, it is not a lower bound or probably even within a factor 3 of the real answer. The limit where QO are observable depends on the noise in the detection system, but $R_D = \exp(-\pi/3)$, or $\omega_c \tau = 3$ is usually quite possible. Furthermore, If we use 26T and 18.5kT we get 1900 A, but we could also use 3kT and 22T, which gives 900 A, highlighting the issues with this estimate. The standard way to determine the mfp is to do a Dingle

analysis. It is not clear why this has not been done here. It was in Ref 33, with mfp determined to be 850 Å.

6) The fit to the zero degree data. The authors claim this shows there is only one frequency. This cannot be true, because if it was, the data would exponentially increase in amplitude with increasing field. What actually happens is that it clearly saturates at high field. The authors should add a plot of amplitude versus field to show this. The fact that the model is unable to resolve the difference simply tell you there are too many free parameters and there is not enough field range. It does not 'reveal the oscillatory contribution of two distinct Fermi surface sections of identical cross-sectional area'.

7) Finally, in the introduction the authors state 'These include a negligible change in the Knight shift upon cooling through T_c as probed by nuclear magnetic resonance (NMR)', for balance the authors should also quote here Matsumura et al [ref 45] which shows the exact opposite. This is references later on but here it should also be discussed as it has a significant bearing on the triplet discussion. I advise the authors to keep an open mind about this. This is important as the history of the literature on Sr₂RuO₄ will attest.

Reviewer #5 (Remarks to the Author):

Eaton and colleagues report quantum oscillation experiments on the candidate spin-triplet superconductor UTe₂, whose properties are currently a very vivid and fast-moving topic in condensed matter physics. Quantum oscillations as a well-established probe for the Fermi surface properties may give important insights relevant to a deeper understanding of the superconductivity in UTe₂.

The manuscript has been reviewed extensively, and the authors have improved their work following the reviewers' comments. The remaining controversy is focused on (i) whether the significance of the results warrants publication in Nature Communications and (ii) whether the claim of the absence of any 3D Fermi surface pockets is supported sufficiently by their data.

The intense research efforts of the community on UTe₂ call for the timely publication of helpful results on the one hand and for carefully avoiding overly bold statements about the significance of the results on the other hand.

Regarding (i) the scope is that "Papers published by Nature Communications aim to represent important advances of significance to specialists within each field." Given the broad interest in UTe₂ and the significance of the normal-state Fermi surface for a deeper understanding, the paper seems suitable for publication in Nature Communications. While the results are not groundbreaking, they represent an important further step.

Regarding (ii) I agree with previous reviewers that the torque data and comparison to the specific heat presented here do not rule out the possible existence of 3D pockets. Rather, the authors present datasets and interpretations that show that their own and other quantum oscillatory data may be explained without invoking the existence of 3D sheets. Not having found convincing evidence for the existence in an extensive dataset is not the same as having proven the absence.

The authors now include some of their new contactless resistance data in the supplementary information and refer to their follow-up manuscript recently posted on the arXiv. This work gives an alternative interpretation for low-frequency oscillations seen in SdH-like data by other groups as well as by the present authors. In fact, if the present manuscript would completely rule out 3D pockets the follow-up work would seem to add nothing to the point. I thus recommend to specify the statement starting with "We therefore identify..." in the sense, that the authors have extensive data and can explain this data without invoking 3D pockets, thus finding no evidence for the

existence of 3D pockets (instead of claiming that there ARE no 3D pockets). After all, this statement relies partly on a different interpretation of their SdH data as compared to [arXiv2303.09050] and is thus open to scientific debate.

A specific statement along these lines would aid the scientific community more in settling the question of the fermiology of UTe₂ by public scientific discourse. With such a change in wording, I can recommend publication

Reviewer responses v3

Reviewer #2 comments:

The authors have addressed all of our outstanding concerns with the manuscript. The addition of an explanation for the relatively low resistivity anisotropy in UTe₂ due to the warping of the cylinders addresses an ongoing question in the field. We feel that the manuscript is now suitable for publication.

A: We thank the reviewer for their thorough reviewing of our manuscript, and for greatly helping us to improve it from the initial draft. We are pleased to hear they find that our addition of the anisotropy analysis “addresses an ongoing question in the field” and thus they feel it “is now suitable for publication.”

Reviewer #4 comments:

The authors report high field torque measurements of the de Haas van Alphen effect in UTe₂. Such measurements are important in order to determine the shape of the Fermi surface and its degree of renormalisation which are themselves important inputs to models to describe the superconductivity in this material. As the authors state, this material has many interesting properties such as very high field re-entrant superconductivity. Many people are working on this compound, so it is highly topical.

In terms of novelty, the data reported here do add to those reported previously by Aoki et al JPSJ 2022 (Ref 33); adding a c→b scan. The angle range where oscillations are detected however, is generally much less than in ref 33 but the current authors do have one single data point at high angle 74 deg, and some at very low angle (<10 deg) which possibly help to constrain their model. This is odd as the present authors state the RRR of their sample is 900 versus 200 in Ref 33 and furthermore their maximum field is much higher, so you would expect large QO and a larger angle range over which they can be observed. The authors do not comment on this (they should). Presumably, it means there is some other damping mechanism at play, but I cannot speculate as to what it might be. Eaton et al also report an effective mass for the overlapping orbits at zero degrees, whereas Aoki et al report orbit specific masses at a higher angle where the different signals can be separated.

The main advance in the submitted m/s is that the authors have suggested a simple model which agrees with the measured angle dependence of the QO, whereas DFT calculations (here and in Ref 33) do not in detail. This has value but given that it is an empirical model (not based on ab-initio calculations) there are some questions about the uniqueness of the solution.

A: We thank the reviewer for their careful reading of our manuscript, for identifying the novelty of our dataset, and for their knowledgeable appraisal of where our work fits within the “highly topical” research landscape.

Specific issues with the paper are

1) In the abstract and elsewhere the authors refer to ‘considerable undulation but negligible small-scale corrugation’. It is not clear to me what this means. This issue is made worse by incomplete details about the empirical model used, and issues with the DFT calculations.

Let me comment on the DFT first. The calculated Fermi surface for the various U values in the SI do not appear to have been calculated with sufficient k-points, and/or an inappropriate interpolation scheme. For example, the blue surface in fig S1 has strange pockets and undulations. This is almost certainly caused by insufficient dense k-mesh. This is further evident for the larger value of U, e.g., S3. The now complete surfaces have 'ripples'. This is a well-known problem which results from using linear cartesian interpolation with a bct k-mesh. The points are not equally spaced in cartesian coordinates and so the interpolation function oscillates. It can be solved, either by reevaluating the energies on an equally spaced cartesian mesh, or using an interpolation scheme which respects the bct symmetry. Increasing the density of the mesh is less effective but does help. Note ref 33 does not have these problems.

For the calculations themselves, the authors should state the structural parameters, including the internal positions (z values), and whether these were relaxed or not. The size of the k-mesh used should also be given in the IBZ (probably a factor 8 lower).

When describing the empirical model, the authors should give further details. They state, 'dominant super-ellipse contribution comes from $n = 5$ ', but what values were used? Presumably a mix of different n values. Were these got from a fit to the DFT? Details should be given.

In general, the DFT actually does a reasonably good job of describing the FS and reproducing the data. However, as the authors note, it does not describe the lower frequency branch of the hole pocket, presumably because the warping of this pocket is less in the DFT than reality. The relation between the empirical model and the DFT needs to be described in more detail. Some suggestions to do this would be (a) show the fitting of the model to the DFT to give the values of n, (b) show how the FS area varies as a function of k_z for the model and the DFT; perhaps pointing out any 'undulations' in the DFT (if they exist) (c) Fit the DFT to the model to show the relationship between the warping parameters. (d) show the positions of the extremal orbits (as a contour line) on the surfaces, so it is clear where the DFT is coming up short.

In reality the model is not simply based on the data, but is constrained by the DFT. They should make this clear. Furthermore, a significant part of the fit constraints do not come from data in the present study, but rather Ref 33.

A1: We thank the reviewer for letting us know that our distinction between undulation and small scale clarification is not clear as this is a key point of our argument about the Fermi surface geometry. When referring to undulation we are referring to the sinusoidal trace that the cylinders make in space where the Fermi surface area remains constant, when we refer to corrugation we are referring to a small-scale neck-and-belly type effect where the cross-sectional area of the Fermi surface. We have produced Supplementary Figure S9 to demonstrate the differing orbits that occur from an undulating cylinder and a corrugated, neck-and-belly cylinder. Most notably, when starting from the c-axis and rotating away, an undulating cylinder has two orbits of similar area, one of which at low angles decreases in area and one which increases. At high angles the areas of both orbits are increasing; this is what we see in our results. For a corrugated neck-and-belly cylinder, on-axis there are two orbits which both increase monotonically as the angle away from the axis increases.

We appreciate, how in the DFT, it could be misconstrued that this corrugation which we have referred to is the ripples from a low density k-point mesh rather than the aforementioned behaviour. In any case, we have redone the calculations on an enhanced k-point mesh of 100,000 points at which this rippling does not occur. Even with this enhanced mesh, the result that the DFT cannot describe this low frequency band holds. To clarify how the DFT was performed we have also now added the internal

coordinates and lattice parameters used to the Supplementary Material, we chose these such that they are the same parameters used as in Ref 33.

One of the key failures of the DFT models are their inability to adequately describe the low frequency branch. This is important since this low frequency branch has a relatively constant frequency over the range that it is observable. If it cannot be described by the two cylinder model, it could suggest that it arises from a 3D pocket. We stress the importance of our empirical model is that it demonstrates conclusively that all the data we have measured are consistent with a two cylinder model. The initial empirical model is of course informed by the DFT in such it predicts the existence of two super-ellipsoidal cylinders, which we then fit to the data. We stress that the exact value of n does not have a major effect on the final results, as the frequency dependencies we find are driven by the undulations rather than exact cross-sectional shape of the Fermi surface. However, we have also provided a fit of n in our empirical model to the DFT Fermi surface for $U = 8.0$ eV in Supplementary Figure S10 that shows that both cylinders are well described by super-ellipses with $n \sim 5$.

A further failing of the DFT is its prediction the existence of multiple frequencies when field is aligned along the c -axis. This occurs due to the corrugation mentioned before. In order to demonstrate this, we have taken the thoughtful advice of the reviewer on board, and as suggested we now plot the cross-sectional area of the sheets as a function of k_z which can now be seen in Supplementary Figure S7. Here we see that all the DFT models have substantially more corrugation than the empirical model, allowing us to achieve a superior fit to the data.

Again we want to thank the reviewer for their advice regarding the modelling process. The inclusion of these features we hope helps to make a much stronger case for the accuracy of our model. We hope that by re-rendering figures S1-S6, and with the addition of three new figures S7, S9 & S10 and their accompanying discussion, that this aspect of our study is now much clearer.

2) Are there any small pockets? Probably not, but the authors have overstated the accuracy to which they can determine this. First of all they assume both surface have the same mass, which does not vary with k_z . In Ref 33 the authors show that the effective masses of the different orbits differ quite a bit, e.g., at 29 degrees 57 versus 33 (difference is 70%). It could be that this just reflects the change in area, but in any case they authors need to quantify the uncertainty from their approximation. Can they quantify any difference in mass by making a 2 mass fit to R_T ? Secondly, their calculation of γ from their own data is possibly suspect. They used the 2D expression for the relationship between γ and the mass, and then scaled this according to the area of their warped cylinder. It is not obvious to me that that would work (but it might). A better way would be to set v_F for their cylinder to match the measured mass, using $m^* \propto dA/dE$, then perform the integration, across the surface.

It is unlikely that after they have properly determined the error in γ , they would be room for a small (say 20-30mJ) contribution from some small 3D pockets. So I don't think they can rule these out.

A2: Regarding the specific aspects raised by this question, we recall our response to Q6 of Reviewer #1 in the first review round: *"In short, we chose to perform the calculation using the mass obtained for field applied parallel to the cylindrical axes because here the dHvA amplitude is considerably larger than at other angles (Fig. 2). This is because collinear to the axes, a large number of k_z states are contributing to the dHvA signal, therefore we are sampling a large proportion of the variation of v_F*

along the cylinders' surfaces. We expected that the two cylinders may exhibit quite different masses for this field orientation, but as we show in Fig 3 a single-component LK fit fits very well, indicating that the masses of both sheets are very similar here. This is consistent with unpublished data reported by D. Aoki at the recent APS March meeting, which showed masses in the range of 30-50 m_e close to the c -axis. Therefore, we are confident that the use of our determined value of 41 m_e is well justified for performing the comparison with the Sommerfeld coefficient. We note that the heavier masses we observe at an inclined tilt angle – at which point the oscillatory amplitude is much smaller as we are only sampling a thin slice of k states – may imply a rather inhomogeneous variation of vF , perhaps especially so at different max/min of the undulations of the cylinders, which will be interesting to follow up in future studies. However, this inhomogeneity is not so pertinent when comparing to the Sommerfeld coefficient – it is the average of vF that is of most importance, and thus sampling the maximal number of k states, where the oscillatory signal is by far the largest, is the best way to achieve this. Hence, this was the procedure we employed.” We hope this covers the similar specific issues raised by Q2 of Reviewer #4 in this third review round.

More broadly, we are very grateful that both Reviewer #4 and Reviewer #5 have advised us to reconsider our wording regarding ruling in/out the possibility of small 3D Fermi surface pockets. We have taken these comments on board, and have significantly revised several sections of the main text to reflect that we cannot rule in/out 3D sections, but instead that our dataset and analysis finds no evidence for the presence of 3D Fermi surface sections. We hope that both reviewers will find this rewording to have improved our manuscript.

3) Zero degrees. The authors state that, for example, the mass measurements are taken at zero degrees. The torque is always zero at this point, so this cannot be. In several places they quote the angle accuracy to be 2 deg, but they say there could be a 5 deg misalignment. They should be consistent. Relative error and systematic error. Did the authors measure both positive and negative angles to show any offset? In the methods they should describe the torque factor ($dF/d\theta$). It would be useful (in the SI) to show amplitude as a function of angle.

A3: We thank the reviewer for raising this point. In our Methods section, we state that “...our angular data obtained in the c – a plane is calibrated to within $\approx 2^\circ$ of experimental uncertainty; however, a possible azimuthal offset in the c – b angles means that these data should only be taken to be accurate to within $\approx 5^\circ$ ” i.e. the two different uncertainties are for the two separate measurements, one performed in one rotation plane and one in the other. We therefore do not feel that this is inconsistent, as the reviewer suggested. Regarding the nature of the torque being zero directly along a high symmetry direction, we added in our prior review rounds a discussion pertaining to this in the supplement: “We note that our employed cantilever beam magnetometry technique of capacitive torque magnetometry in general allows for better accuracy and precision in angular orientation than magnetic torque performed using piezoelectric cantilevers, due to the larger sample and cantilever sizes involved that enable easier orientation during the mounting procedure. However, by the nature of the measurement, the possibility of some non-negligible deflection of the cantilever away from the equilibrium position of alignment – although unlikely due to the very small magnitude of background torque close to a high symmetry direction – nonetheless cannot be entirely excluded.” Furthermore, in Fig. 3 of the main text we write $H \sim c$ rather than $H//c$ to account for the fact that this is of course not precisely perfectly along c , because as the reviewer correctly notes a totally perfect alignment (which of course is almost experimentally impossible) would result in zero net torque, even in the high field range of our experiment. We hope that the reviewer finds this discussion and choice of nomenclature satisfactory.

4) One useful part of the current work is the fact they found no additional orbits. Interestingly they do find an orbit at very high angle (74 deg). However, they do not state what angle range and spacing they searched (e.g. full 90 deg with 5 deg spacing?). I suspect that the one point they find at very high angle is in fact the so called Yamaji point, where the amplitude is massively enhanced. This would throw into question their model and the correspondence between this point and the re-entrant superconductivity. Is this a singular point where QO were observed? This needs to be discussed. Furthermore, they state that 'This implies that a sharp peak in the density of states may underpin the microscopic mechanism driving this exotic superconducting state.'. The density of states is obviously not angle dependent, so it is unclear what they mean by this. The Yamaji point is actually where A_k does not vary with k_{\parallel} , or there is a peak in the so-called curvature factor.

A4: We welcome the reviewer's comments on this specific aspect of our article. Unlike ref. 33, in which all the measurements were taken in a home lab, all of our dHvA measurements were obtained at the National High Magnetic Field Laboratory, Tallahassee, USA. Use of magnet time at user facilities is highly valuable, and competitively applied for and allocated. Therefore, unlike the study of ref. 33 we had only a limited amount of time in which to perform our investigation. Thus, in the c-a rotation plane we sought to replicate some of the angle points previously reported by ref. 33, which we found to be well reproduced by our measurements. We then deliberately focused on obtaining data close to the c direction, and far away from c, as these extrema are most useful for constraining the Fermi surface geometry, and were not reported in ref. 33. Due to the time pressures of magnet time we did not perform measurements every 5 degree steps, as the reviewer presumed, but instead deliberately invested considerable time to obtaining the oscillatory frequency at a high inclination angle (which happened to be 74deg) at which point the signal was very small and we thus had to sweep the field very slowly (0.05 T/min as we specify in the Supplement) and average over multiple sweeps to resolve the faint, fast signal. This therefore bears no relation to the discussion of the Yamaji point, which incidentally is only pertinent to the magnetic field reentrant superconductivity in the c-b rotation plane, not for c-a. We hope that this explanation satisfies the reviewer's question of this aspect of our dataset.

5) Determination of the mean free path. The authors have attempted to determine the mean free path by equating the cyclotron radius with the mean free path. This is equivalent to setting $\omega_c\tau=1$ at this field, so $R_D=\exp(-\pi)$. Although this is called 'approximate' there is no discussion about how 'approximate'. It is further stated that this is a 'lower bound'. Although I agree this gives an order of magnitude estimate, it is not a lower bound or probably even within a factor 3 of the real answer. The limit where QO are observable depends on the noise in the detection system, but $R_D=\exp(-\pi*3)$, or $\omega_c\tau=3$ is usually quite possible. Furthermore, if we use 26T and 18.5kT we get 1900 A, but we could also use 3kT and 22T, which gives 900 A, highlighting the issues with this estimate. The standard way to determine the mfp is to do a Dingle analysis. It is not clear why this has not been done here. It was in Ref 33, with mfp determined to be 850 A.

A5: We thank the reviewer for pointing out the fact that using the values of 18.5kT and 26T one obtains the value of 1900A, which is the entirety of the claim we are making here. As the observation of 18.5kT for one orientation and 3kT at another were made on the same sample, the appropriate lower bound to take for the sample's mfp is thus naturally the higher one (i.e. 1900A). As the reviewer correctly identifies, noise in the detection system can make high frequency small amplitude oscillations difficult to resolve – but this then leads to the highest observed frequency being a lower bound for identifying the mfp (as it could be the case that the mfp is in fact higher, with higher frequency oscillations being hidden by the noise floor). These arguments are why we have described this as a lower bound.

Regarding the question of a Dingle analysis, we recall our response to Q14 from Reviewer #2 in the first round of referee reports: *“The reviewer raises a good point, as the Dingle term is a standard method for estimating the mean free path. However, to calculate it properly, one needs to know both the effective mass from dHvA and the band mass. However, as we go to great lengths to show in our supplement, DFT calculations of UTe₂ fail to properly capture the electronic structure of this material, so we do not have properly deduced band mass values with which to perform the calculation. Therefore, we elected not to include a Dingle calculation, and instead include other estimates of the mean free path in our supplement, calculated from data we collected that are of high accuracy and precision, yielding estimates for the mean free path that are in good agreement with each other. We feel that this is sufficient, and that there would be no gain from making a crude approximation of the Dingle term from incorrectly deduced band masses.”* We hope that Reviewer #4 agrees with Reviewer #2 that this concise consideration of the mean free path is adequate – further, we note that this supplementary section is not of direct importance or relevance to any of the results we draw from our study, but was merely included for completeness for interested readers.

6) The fit to the zero degree data. The authors claim this shows there is only one frequency. This cannot be true, because if it was, the data would exponentially increase in amplitude with increasing field. What actually happens is that it clearly saturates at high field. The authors should add a plot of amplitude versus field to show this. The fact that the model is unable to resolve the difference simply tell you there are too many free parameters and there is not enough field range. It does not ‘reveal the oscillatory contribution of two distinct Fermi surface sections of identical cross-sectional area’.

A6: We are grateful that the reviewer raised this point, as it echoes Q1 of Reviewer #1 in the first round of referee questions. We recall our response: *“We thank the reviewer for their careful inspection of our manuscript. Regarding this small feature of our dataset, we regret that this is very likely a spurious artifact of the dilution fridge’s temperature stability. Our experimental procedure for measuring each angle point was to sweep the field down to 1 T, where the Hall sensor can obtain a good reading. The magnet was then swept back up to 28 T at a comparatively fast sweep rate (0.5 T/min), taking almost an hour. An effect of this faster sweep rate is that eddy currents in the mixing chamber raise the temperature from base (~0.02 K) up to ~0.1 K. After returning to 28 T, we then waited typically another hour for the cooling power of the fridge to dissipate this additional heat load. Unfortunately, in the instance mentioned by the reviewer, some cooling was very likely still occurring over the field interval of approximately 27-28 T. Due to the heavy effective mass, this regrettably leads to a slight extraneous diminution of the oscillatory amplitude. Having unfortunately only realised this after the measurement, in performing our Lifshitz-Kosevich analysis of the temperature-dependent oscillatory amplitude we were very careful to select the field range of 22-26.5 T in Fig 3, to ensure that the sample and mixture (and thermometer) were in thermal equilibrium, to give reliable results.”* As this small aspect clearly still confuses close readers of our manuscript, we have added some discussion of this to the Supplementary Text. We note that at fields up to 26.5T the amplitude does indeed rise as the reviewer correctly identifies would be consistent with our interpretation. Furthermore, the fitting of the α and β components clearly indicates the uncertainties (15 T and 13 T, respectively) identified by this procedure. We hope that by adding this additional discussion, this small aspect of our dataset will not cause confusion for any future readers.

7) Finally, in the introduction the authors state ‘These include a negligible change in the Knight shift upon cooling through T_c as probed by nuclear magnetic resonance (NMR)’, for balance the authors

should also quote here Matsumura et al [ref 45] which shows the exact opposite. This is references later on but here it should also be discussed as it has a significant bearing on the triplet discussion. I advise the authors to keep an open mind about this. This is important as the history of the literature on Sr₂RuO₄ will attest.

A7: We thank the reviewer for raising this important point. Indeed the history of Sr₂RuO₄ is important to bear in mind when embarking on studies of UTe₂. In our revised manuscript we have added a reference not only to the new NMR data obtained on MSF UTe₂ samples, but also to the recent thermal conductivity, Kerr effect and muon spin rotation reports, each of which provide a contrasting view to the prior interpretation of studies on CVT-grown samples. We are grateful to the reviewer for prompting us to add this additional discussion to improve our manuscript.

Reviewer #5 comments:

Eaton and colleagues report quantum oscillation experiments on the candidate spin-triplet superconductor UTe₂, whose properties are currently a very vivid and fast-moving topic in condensed matter physics. Quantum oscillations as a well-established probe for the Fermi surface properties may give important insights relevant to a deeper understanding of the superconductivity in UTe₂.

The manuscript has been reviewed extensively, and the authors have improved their work following the reviewers' comments. The remaining controversy is focused on (i) whether the significance of the results warrants publication in Nature Communications and (ii) whether the claim of the absence of any 3D Fermi surface pockets is supported sufficiently by their data.

The intense research efforts of the community on UTe₂ call for the timely publication of helpful results on the one hand and for carefully avoiding overly bold statements about the significance of the results on the other hand.

Regarding (i) the scope is that "Papers published by Nature Communications aim to represent important advances of significance to specialists within each field." Given the broad interest in UTe₂ and the significance of the normal-state Fermi surface for a deeper understanding, the paper seems suitable for publication in Nature Communications. While the results are not groundbreaking, they represent an important further step.

Regarding (ii) I agree with previous reviewers that the torque data and comparison to the specific heat presented here do not rule out the possible existence of 3D pockets. Rather, the authors present datasets and interpretations that show that their own and other quantum oscillatory data may be explained without invoking the existence of 3D sheets. Not having found convincing evidence for the existence in an extensive dataset is not the same as having proven the absence.

The authors now include some of their new contactless resistance data in the supplementary information and refer to their follow-up manuscript recently posted on the arXiv. This work gives an alternative interpretation for low-frequency oscillations seen in SdH-like data by other groups as well as by the present authors. In fact, if the present manuscript would completely rule out 3D pockets the follow-up work would seem to add nothing to the point. I thus recommend to specify the statement starting with "We therefore identify..." in the sense, that the authors have extensive data and can explain this data without invoking 3D pockets, thus finding no evidence for the existence of 3D pockets (instead of claiming that there ARE no 3D pockets). After all, this statement relies partly on a different interpretation of their SdH data as compared to [arXiv2303.09050] and is thus open to scientific debate.

A specific statement along these lines would aid the scientific community more in settling the question of the fermiology of UTe₂ by public scientific discourse. With such a change in wording, I can recommend publication

A: We sincerely thank the reviewer for their detailed reading of our manuscript and previous referee replies. We gratefully acknowledge their astute summary of the current status of the subfield of UTe₂ research and where our work falls within in. We thank the reviewer for their very well balanced comments, and we agree that “Given the broad interest in UTe₂ and the significance of the normal-state Fermi surface for a deeper understanding, the paper seems suitable for publication in Nature Communications.”

Regarding the phrasing of the presence of 3D sections, we thank the reviewer for their thoughtful suggestion of how to amend our wording. In our revised manuscript we have completely rewritten the sentence that previously began “We therefore identify...” to instead read that we find “...no evidence indicating the presence of 3D sections in the Fermi surface of UTe₂...” which we hope the reviewer finds to be a much improved choice of phrasing.

In the spirit that this matter of contention should be addressed by “public scientific discourse”, we have also added a penultimate paragraph (immediately preceding the conclusion) discussing our interpretation of oscillatory features in the magnetoconductance of UTe₂ (with reference to recent unpublished work by Dai Aoki, who has also failed to find dHvA signatures of 3D sections despite intensive searching). Furthermore, we have amended the conclusion itself to once more include the reviewer’s suggested phrasing of “...we find no evidence for the presence of any 3D sections.” We hope that having made these amendments, the reviewer now feels that our manuscript is ready for publication.

REVIEWERS' COMMENTS

Reviewer #4 (Remarks to the Author):

The authors have made a number of changes and have improved the manuscript. However, there remain a few issues, which although are small and easy to fix, should be attended to before the manuscript is published. For the benefit of the authors, they should be aware that when reviewing the manuscript I was only given the reports/responses of the previous round of review, so not the first round which they refer to in their reply. However, since I found some of the same issues, this suggests that the changes made previously were not sufficient to answer the questions. The authors should check this information is included in the manuscript / SI, as not all readers will read the reviewers file.

- 1) DFT calculations. The authors have provided the necessary details of structure used, and have re-calculated the surfaces with a finer mesh. This answers the issues raised here.
- 2) Meaning of undulation and corrugations. This has been answered by the new figure added in the SI.
- 3) Small pockets. The authors have answered this only in part. Since this is a crucial question that many readers will be interested in, it is vital this is answered completely and accurately. First of all, although in one part of the manuscript they have modified the text to say simply that they see no evidence for 3D pockets, in another part they say 'makes the presence of any 3D Fermi surface pocket(s) extremely unlikely', which is a much stronger statement (not modified from the original).

Later in the manuscript the authors state 'Given the close correspondence between the values of γ_N measured by specific heat experiments and calculated from our dHvA data and Fermi surface simulations, this adds strong confidence to our Fermi surface simulations and interpretation of the dHvA data that these two quasi-2D sections likely comprise the only Fermi surface sheets present in UTe₂'

Although it is true that the calculated 120.5 mJ/mol/K² is very close to the measured 121(1) mJ/mol/K², the former number does not include an associated error. At the very least this should be 5% (2/41, the error in the mass), but I think it should be actually larger if the authors also account for the error introduced by assuming the masses of both sheets are the same and do not vary with k_z . For example, In figure 3c, they show the fit to determine the mass. This has only 5 data points (the field range is not stated nor is the uncertainty in the temperature measurements), and is fit to a single component. However, it is possible to fit the same data to two components with 2 masses -e.g, $1.5 \cdot m_{\text{mean}}$ and $0.5 \cdot m_{\text{mean}}$. The fit quality improves but now there are 3 free parameters instead of two. This is a physically plausible situation and would give a different value of γ if the two masses are summed with a weighting according to their surface area. This possibility needs to be taken into account when calculating the error, as well as the temperature uncertainty. It is only if the calculated γ agrees with the measured one within the error of both that we can form the conclusion the authors have given.

Another source of error is the assumption that v_F does not vary with k_z . The authors note that their results and those from Aoki et al 'may imply a rather inhomogeneous variation of v_F '. Although, I agree with the authors that the number of k -states contributing to the orbit will vary with angle, it is not at all obvious that at (say) 30degrees the difference would be large compared to zero degrees. Calculations are necessary to show this, but none are presented. It should be straightforward to calculate the angle dependence of the effective mass for their model and hence compare to the data from Aoki et al. If it does not agree, the authors should state this. It would help to quantify the accuracy of their model, which is important for answering the question about the existence or not of 3D pockets.

- 4) Zero degrees. Although in figure 3 $H \sim C$ is written, this is not repeated in figure 2, or in the text, 'raw torque signal at 0^{deg}'. It would be better to add, 'approximately' or 0 ± 2 degrees. Furthermore, the authors have not changed the text in the manuscript which I feel is still misleading.

'Hence, one observes a large quantum oscillatory amplitude for magnetic field oriented in this direction (in this case the c direction). Then, as the field is tilted away from the axis of the cylinder, the oscillatory amplitude falls considerably due to phase smearing that increases with the rate of change of the frequency with angle'

Although this is true if simple magnetisation were measured, this is not true for the torque signal which was measured here. The signal is exactly zero for $H//c$, not maximal. The signal is maximal at a finite angle which depends on the variation of the torque factor and the mean-free-path. This text should be revised.

I previously suggested the authors 'In the methods they should describe the torque factor (dF/θ) '. They have not done this or replied to this.

Reviewer #5 (Remarks to the Author):

The authors have addressed the points raised by me appropriately. I recommend publication.

Referee responses v4

Reviewer #4 (Remarks to the Author):

The authors have made a number of changes and have improved the manuscript. However, there remain a few issues, which although are small and easy to fix, should be attended to before the manuscript is published. For the benefit of the authors, they should be aware that when reviewing the manuscript I was only given the reports/responses of the previous round of review, so not the first round which they refer to in their reply. However, since I found some of the same issues, this suggests that the changes made previously were not sufficient to answer the questions. The authors should check this information is include in the manuscript / SI, as not all readers will read the reviewers file.

1) DFT calculations. The authors have provided the necessary details of structure used, and have re-calculated the surfaces with a finer mesh. This answers the issues raised here.

A1: We are grateful for the reviewer's careful reading of our manuscript, and for their constructive criticism of how to improve it. We are pleased to hear that the reviewer is satisfied with our updated DFT supplementary section.

2) Meaning of undulation and corrugations. This has been answered by the new figure added in the SI.

We are glad that the reviewer finds the inclusion of this new figure to aid the clarity of our manuscript.

3) Small pockets. The authors have answered this only in part. Since this is a crucial question that many readers will be interested in, it is vital this is answered completely and accurately.

First of all, although in one part of the manuscript they have modified the text to say simply that they see no evidence for 3D pockets, in another part they say 'makes the presence of any 3D Fermi surface pocket(s) extremely unlikely', which is a much stronger statement (not modified from the original).

Later in the manuscript the authors state 'Given the close correspondence between the values of γ_N measured by specific heat experiments and calculated from our dHvA data and Fermi surface simulations, this adds strong confidence to our Fermi surface simulations and interpretation of the dHvA data that these two quasi-2D sections likely comprise the only Fermi surface sheets present in UTe2'

Although it is true that the calculated 120.5 mJ/mol/K^2 is very close to the measured $121(1) \text{ mJ/mol/K}^2$, the former number does not include an associated error. At the very least this should be 5% ($2/41$, the error in the mass), but I think it should be actually larger if the authors also account for the error introduced by assuming the masses of both sheets are the same and do not vary with k_z . For example, In figure 3c, they show the fit to determine the mass. This has only 5 data points (the field range is not stated nor is the uncertainty in the temperature measurements), and is fit to a single component. However, it is possible to fit the same data to two components with 2 masses -e.g,

$1.5 \cdot m_{\text{mean}}$ and $0.5 \cdot m_{\text{mean}}$. The fit quality improves but now there are 3 free parameters instead of two. This is a physically plausible situation and would give a different value of γ if the two masses are summed with a weighting according to their surface area. This possibility needs to be taken into account when calculating the error, as well as the temperature uncertainty. It is only if the calculated γ agrees with the measured one within the error of both that we can form the conclusion the authors have given.

Another source of error is the assumption that v_F does not vary with k_z . The authors note that their results and those from Aoki et al 'may imply a rather inhomogeneous variation of v_F '. Although, I agree with the authors that the number of k -states contributing to the orbit will vary with angle, it is not at all obvious that at (say) 30 degrees the difference would be large compared to zero degrees. Calculations are necessary to show this, but none are presented. It should be straightforward to calculate the angle dependence of the effective mass for their model and hence compare to the data from Aoki et al. If it does not agree, the authors should state this. It would help to quantify the accuracy of their model, which is important for answering the question about the existence or not of 3D pockets.

A3: We thank the reviewer for raising a number of good points, and for suggesting how we could improve this aspect of our article. Regarding the wording of the likelihood of 3D sections, we note that our phrasing here was previously found satisfactory to Reviewer #5 – however, in our latest version we have now modified this to be less strong, which we hope will also now be suitable for Reviewer #4. Regarding the possibility of fitting to two separate masses for the LK analysis, and the uncertainty in the evolution of the effective mass with angle for different slices of the Fermi surface cylinders, the reviewer raises some excellent points about comparing the effects of over parameterisation with lower uncertainty – we have added some discussion to the methods to highlight this uncertainty, and to suggest how future measurements could seek to reduce this uncertainty. Regarding the reviewer's suggestion of predicting the angle dependence of the effective mass for our model, while we would very much like to do this, as our model is simply formed from geometric considerations, the dominant role of strong correlations giving rise to highly renormalised carrier masses (by ~an order of magnitude from simple DFT calculations) unfortunately renders this not possible. We note further that a unique aspect of our model is that down the c -axis the entire Fermi surface is extremal and thus contributing to the observed signal (including light and heavy sections). We regret not highlighting this before, as it has likely been a source of confusion. We have now emphasised this point in our revised manuscript, which we hope will greatly assist readers in understanding our calculation of the k -dependent density of states.

4) Zero degrees. Although in figure 3 $H \sim c$ is written, this is not repeated in figure 2, or in the text, 'raw torque signal at 0° '. It would be better to add, 'approximately' or 0 ± 2 degrees. Furthermore, the authors have not changed the text in the manuscript which I feel is still misleading. 'Hence, one observes a large quantum oscillatory amplitude for magnetic field oriented in this direction (in this case the c direction). Then, as the field is tilted away from the axis of the cylinder, the oscillatory amplitude falls considerably due to phase smearing that increases with the rate of change of the frequency with angle' Although this is true if simple magnetisation were measured, this is not true for the torque signal which was measured here. The signal is exactly zero for $H // c$, not maximal. The signal is maximal at a finite angle which depends on the variation of the torque factor and the mean-free-path. This text should be revised. I previously suggested the authors 'In the methods they should describe the torque factor ($dF/d\theta$)'. They have not done this or replied to this.

A4: We thank the reviewer for raising these points. The reviewer is correct in stating that the torque is minimised along a high symmetry direction. While this is in general true for the background torque (that we call τ), the case is more subtle for the oscillatory component (that we call $\Delta \tau$), which if there was a non-zero background amplitude would be maximal along symmetry for e.g. a cylindrical Fermi surface. However, we feel that this is naturally understood by our statement of a small angular uncertainty in the Methods, and we do not feel this small point meaningfully alters our interpretation of the data in any way. However, it is of course preferable to avoid any doubt whatsoever. One of the reasons we included panel 2c was to highlight that, for 0deg and 74deg, although the background torque of 0deg is less than that at 74deg (as the reviewer correctly identifies that one would expect), the oscillatory amplitude is clearly very different, with 0deg $\Delta \tau$ over an order of magnitude higher than at 74deg. To address this common misconception regarding torque magnetometry experiments, which may also confuse some readers, we have added some additional phrasing to the Supplementary Information discussing this. We hope that this adds greater clarity to this subtle aspect of our measurement technique, and we thank the reviewer again for inviting us to improve this aspect of our article.

Reviewer #5 (Remarks to the Author):

The authors have addressed the points raised by me appropriately. I recommend publication.

We thank the reviewer for their constructive feedback on improving our manuscript.